# Crowding-induced opening of the mechanosensitive Piezo1 channel in silico

Wenjuan Jiang[1], John Smith Del Rosario [2,4], Wesley Botello-Smith[1,4], Siyuan Zhao[2], Yi-chun Lin[1], Han Zhang[1], Jérôme Lacroix [3,5 ✉], Tibor Rohacs [2,5 ✉] & Yun Lyna Luo [1,5 ✉]

Mechanosensitive Piezo1 channels are essential mechanotransduction proteins in eukaryotes. Their curved transmembrane domains, called arms, create a convex membrane deformation, or footprint, which is predicted to flatten in response to increased membrane tension. Here, using a hyperbolic tangent model, we show that, due to the intrinsic bending rigidity of the membrane, the overlap of neighboring Piezo1 footprints produces a flattening of the Piezo1 footprints and arms. Multiple all-atom molecular dynamics simulations of Piezo1 further reveal that this tension-independent flattening is accompanied by gating motions that open an activation gate in the pore. This open state recapitulates experimentally obtained ionic selectivity, unitary conductance, and mutant phenotypes. Tracking ion permeation along the open pore reveals the presence of intracellular and extracellular fenestrations acting as cation-selective sites. Simulations also reveal multiple potential binding sites for phosphatidylinositol 4,5-bisphosphate. We propose that the overlap of Piezo channel footprints may act as a cooperative mechanism to regulate channel activity.

---

[1] College of Pharmacy, Western University of Health Sciences, Pomona, CA 91766, USA. [2] Department of Pharmacology, Physiology and Neuroscience, Rutgers, New Jersey Medical School, Newark, NJ 07103, USA. [3] Graduate College of Biomedical Sciences, Western University of Health Sciences, Pomona, CA 91766, USA. [4] These authors contributed equally: John Smith Del Rosario, Wesley Botello-Smith. [5] These authors jointly supervised this work: Jérôme Lacroix, Tibor Rohacs, Yun Lyna Luo. ✉email: jlacroix@westernu.edu; rohacsti@njms.rutgers.edu; luoy@westernu.edu

Piezos are homotrimeric mechanosensitive channels expressed in many eukaryotic cells. In vertebrates, two isoforms, Piezo1 and Piezo2, transduce various forms of mechanical stimuli into electrochemical signals that contribute to many biological functions, including somatovisceral sensation, proprioception, vascular development, blood pressure regulation, osmotic homeostasis, and epithelial growth[1]. Patients carrying gain or loss of function Piezo mutations present various disease conditions, such as xerocytosis, arthrogryposis, loss of proprioception, and lymphedema[2]. Upregulation of Piezo2 activity correlates with inflammation-induced pain states[3], whereas gain-of-function Piezo1 variants confer Malaria resistance in humans and animal models[4].

Piezo channels sense mechanical cues transmitted directly from the membrane, and thus obey the so-called force-from-lipid paradigm. High-resolution cryo-electron microscopy (cryo-EM) structures of Piezo1 and Piezo2 reveal a unique molecular architecture consisting of three long peripheral transmembrane domains or arms, and a central region harboring a unique transmembrane pore and an extracellular (EC) cap domain[5–8]. Compared with the Piezo2 structure, the upper part of the Piezo1 pore is more dilated, rendering a funnel-shaped pore with a minimum radius at the lower constriction site. A previous simulation showed that this lower constriction site is too narrow to conduct ion and water[9], indicating Piezo1 structures are nonconducting. In addition, the peripheral arms are arranged in spirals, giving Piezos a triskelion shape when viewed perpendicularly to the membrane plane and a bowl-like shape when viewed parallel to it. This bowl shape imposes a local convex curvature, or inverted dome, to the lipid bilayer, suggesting Piezo channels become flatter, and potentially open, when membrane tension increases[6,10,11].

The formation of Piezo1 clusters has been reported by independent studies, which suggest Piezo1 clustering plays a role in concerted gating transitions, such as collective loss of inactivation[12–14]. Hence, the gating properties of Piezo channels may be modulated by the local channel density at the plasma membrane. Due to the intrinsic membrane bending rigidity, the membrane deformation induced by a single Piezo1 channel, or membrane footprint, is predicted to extend well beyond the local dome and decay within tens of nanometers[10]. Simulating Piezo1 channel in such a large membrane is feasible using coarse-grained (CG) molecular dynamics (MD) simulations, but remains challenging for all-atom (AA) MD simulations without sacrificing temporal sampling due to a large number of atoms. In most MD simulations with explicit solvent, the periodic boundary conditions (PBC) create an infinite lattice where the simulated system is infinitely replicated throughout virtual space. Thus, when Piezo1 is simulated in a membrane area smaller than its membrane footprint, the PBC creates a virtual cluster of channels with overlapping footprints between neighboring images.

Here, using a hyperbolic tangent model, we first propose that the overlap of neighboring Piezo1 membrane footprints create membrane topology constraints that flatten the Piezo1 dome, facilitating Piezo1 opening. We tested this hypothesis using AA MD simulations of a crowded Piezo1 cluster with different degrees of footprint overlap. Given the importance of phosphoinositides to Piezo function[15,16], we included 5% phosphatidylinositol 4,5-bisphosphate (PIP$_2$) in the inner leaflet of the 1-palmitoyl-2-oleoy phosphatidylcholine (POPC) bilayer to assess potential lipid–protein interaction during both CG and AA simulations. In good agreement with the hyperbolic tangent model, footprint overlap reduces membrane curvature, leading to a tension-free flattening of the Piezo1 arms and concomitant widening of the pore constriction site within 2 μs AA simulation. The conducting state generated from MD simulation reproduces

ionic conductance and selectivity properties consistent with numerous experimental studies. In addition, we observed consistent PIP$_2$ binding hotspots from multiple simulations and investigated the functional role of these putative binding sites, using functional assays.

## Results

**Hyperbolic tangent model suggests large footprint overlap flattens the Piezo dome.** Using total internal reflection fluorescence (TIRF) microscopy, we found that GFP-tagged Piezo1 channels heterologously expressed in HEK293 cells exhibit a punctate distribution (Fig. 1a), consistent with earlier studies[12–14]. The distribution of Piezo1–GFP puncta fluorescence intensity shows a main peak with secondary peaks at higher values, suggesting puncta correspond to clusters having a varying number of channels (Fig. S1). Within a cluster, the promiscuity of channels footprints may affect membrane geometry and, hence, channel function. To explore this effect of footprint overlap, we first used a hyperbolic tangent model to mimic the overlap of Piezo's membrane footprint. It is expected that the membrane approaches a flat plateau when distant from the Piezo dome (although ripple-like oscillations/perturbations inherent to such membranes prevent this from being strictly monotonic). This property is captured well by the hyperbolic tangent function, which monotonically approaches unity as it moves away from the origin. The ratio of the distance between two Piezo domes, $R$, and the distance from the Piezo dome to the bulk membrane midplane, $H$, acts as a dimensionless separation coefficient $D$ (note: $D = R/H$) for the hyperbolic tangent model (Fig. 1b). Note $R$ is only the distance between two domes (i.e., dome–dome distance), independent of the dome's size. If we define the inclination angle $\alpha$ between the Piezo arm and the $xy$ plane (i.e., membrane plane), the slope of the Piezo dome is $\tan^{-1}(\alpha)$.

This model predicts a biphasic relationship between the footprint overlap angle $\theta$ (larger $\theta$ means flatter footprint overlap) and the inclination of the Piezo1 arm (smaller $\alpha$ means flatter dome; Fig. 1b). When the dome angle $\alpha$ is smaller than a critical value, the footprint flattening (increasing $\theta$) leads to a flattening of the Piezo dome and arms (decreased $\alpha$). In contrast, when the dome angle is larger than this critical angle, the overlap flattening leads to more pronounced arm curvature (increased $\alpha$). Thus, the hyperbolic tangent model suggests that when two Piezo footprints overlap, larger Piezo–Piezo distances favor arm curvature, while smaller Piezo–Piezo distances favor arm flattening. This qualitative biphasic relationship is independent of membrane mechanical properties. However, the exact value of this critical distance (separation coefficient $D$) depends on the geometric parameters of the footprint and bilayer rigidity. Below we show how this model corresponds to the parameters in MD simulations of Piezo1.

**Crowding induces Piezo1 pore opening in three MD simulations.** The hyperbolic tangent model suggests that the overlap of neighboring Piezo1 footprints favors flattening of the membrane at a sufficiently small Piezo dome–dome distance. We hence explore the influence of the footprint overlapping on Piezo conformation using AA MD simulations with PBC. The atomic model was built using a cryo-EM structure of mouse Piezo1 in a nonconducting state at 3.8 Å resolution (PDB ID 6B3R), solved in the detergent micelle in the absence of mechanical stimuli. Piezo1 channel is a large homotrimer with 38 predicted transmembrane helices (TM) per subunit. In this cryo-EM structure, TM1–12 are not present, and TM13–16 side chains are not fully solved. To avoid those structural uncertainties in the atomistic model, we only include TM17–38 for each subunit. Missing loops larger

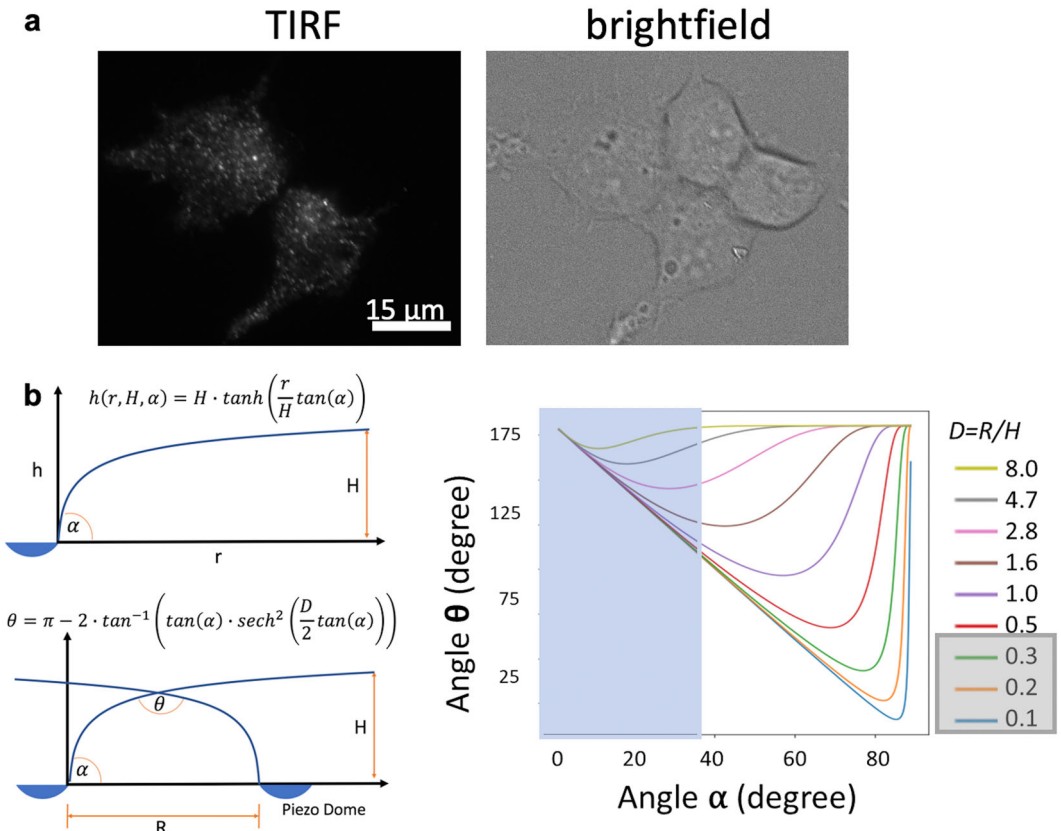

**Fig. 1 Effect of clustering on Piezo1 footprint topology. a** TIRF imaging of HEK293 cells expressing an N-terminal GFP-tagged mPiezo1 shows a punctate distribution. **b** Hyperbolic tangent model, in which $H$ is the distance from Piezo dome to the bulk membrane midplane; $R$ is the distance between two Piezo dome perimeters; $D$ is the separation ratio $R/H$; $\alpha$ is the angle of Piezo1 arm inclination; $\theta$ is the angle of membrane footprint intersection (see "Methods" for details). The biphasic relationship between $\alpha$ and $\theta$, and the dependence on the Piezo separation ratio $D$. For **b**, the blue rectangle overlay depicts the possible range of elevation angles in Piezo1 and the $D$ ratios highlighted in gray show the range $R/H$ values simulated.

than 20 amino acids were not modeled (see "Methods" and "Discussion" sections). Although the arm length in this model is presumably 5/9 of the full length, our simulations show that this model encompasses key structural features, allowing Piezo1 to sense the membrane topology and open its pore.

Prior to AA simulations, a 12 μs CG MD simulation of Piezo1 embedded in a solvated POPC/PIP$_2$ bilayer was first conducted to accelerate the convergence of lipid lateral diffusion (Fig. S2) and dome formation, while the Piezo1 backbone was kept rigid (Table S1). The equilibrated CG system was then converted to an AA system. Three replicas of ~200 ns AA MD simulations without restraint on protein were then conducted with different initial velocities. After equilibration, each replica was truncated into a smaller PBC box with similar membrane surface area, labeled *box1*, *box2*, and *box3*, for production run with optimal performance on the Anton2 supercomputer (Table S2). Each system ran for 1.75 μs before analysis. System *box1* was extended to 2 μs before ionic conductance measurement.

The root-mean-square-deviation (RMSD) of the Piezo1 backbone plateaued after ~1 μs in all systems (Fig. 2a). By subtracting the lipid surface area from the total PBC box *xy* area, Piezo1 occupies 14.8 ± 0.9% total area in the outer membrane and 19.8 ± 0.9% in the inner membrane (Table S2). Because the threefold symmetry is not imposed during MD simulation, we calculate Piezo dome radius ($r$) using an average centroid distance of a triangle defined by the center of mass of the outer helix in each arm (see orange triangle in Fig. 2b). The dome–dome distance ($R$) is hence the box dimension in *x* or *y* minus 2*r*. The smallest dome–dome distance in each simulation is 1–3 nm along the *x*-

dimension and 4–7 nm along *y*-dimension (Table S3). Hence, in our simulations, the Piezo1 membrane footprint of tens of nanometers[10] largely overlaps with the footprint of the Piezo1 mirror images (Fig. 2b, c). Note that dome–dome distance is smaller than protein–protein distance and this virtual cluster does not replicate conformational heterogeneity seen in a real cluster (see "Discussion" section). Nevertheless, this footprint overlap between periodic images creates a considerable energy penalty for membrane deformation in the overlapping region.

To correlate with the hyperbolic tangent model above, we calculated the angle $\alpha$ from Piezo1 cryo-EM structure to be 38°. According to Fig. 1b, this angle corresponds to the critical separation coefficient ($D$) 1.6–2.8. If we borrow the height of the footprint ($H$) for a single Piezo1 from ref. 10, it is on the magnitude of ~10 nm. Using these values, the critical distance $R$ is estimated to be 16–28 nm ($R = D*H$; Fig. 1b), suggesting Piezo1 dome–dome distance <16 nm will reduce membrane curvature. In current MD simulations, the closest dome–dome distance (1–3 nm) is much smaller than 16 nm. Thus, it is guaranteed to reduce the membrane curvature. In fact, our simulated Piezo1 cluster likely represents the lower limit of Piezo dome separation (i.e., highest cluster density). The physiologically relevant $D$ value will depend on the density of Piezo1 clusters on different cell types, and the exact density is yet to be determined.

This hypothesis was validated by fitting the Cartesian coordinates of lipid headgroups into a 2D Gaussian model (see details and code in "Method" section) during the 1.75 μs simulation. This fitting approach revealed a reduction of membrane curvature in all three replicas (Fig. 2c, d). In the

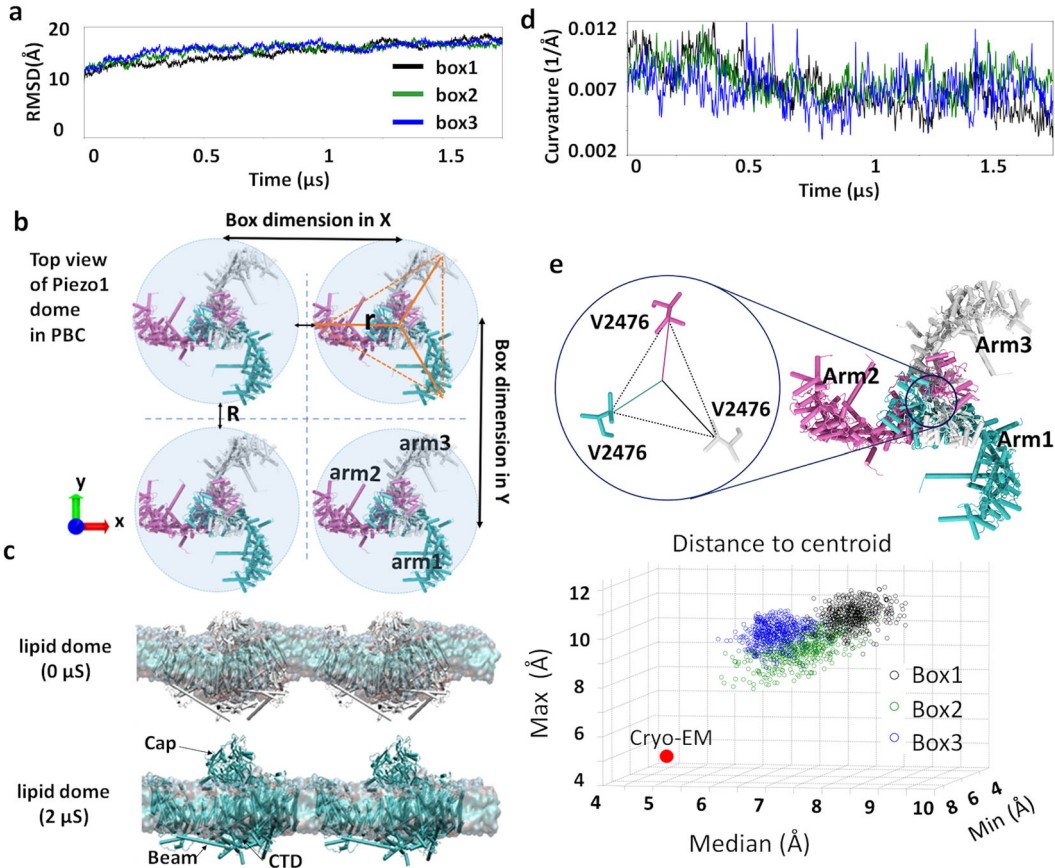

**Fig. 2 Three 1.75 µs Piezo1 all-atom MD simulations. a** Protein backbone RMSD in three systems, named *box1*, *box2*, and *box3*. **b** Piezo PBC images along *x*-axis and *y*-axis. Arm 1 in cyan, arm2 in mauve, and arm3 in white. The centroid distance used to calculate the radius of the Piezo1 dome (*r*) is illustrated using solid orange lines. The dome–dome distance (*R*) and the box size in *x*- and *y*-dimensions are shown using black double arrow lines. **c** Flattening of the lipid dome between two Piezo1 channels, illustrated by the snapshots at 0 and 2 µs of AA simulation of the *box1* system. **d** Mean curvature of three systems from fitting lipid headgroups to a 2D Gaussian model (see "Methods" section for details and code). **e** Scatter plot of three V2476 centroid distances (min, median, and max). The circle symbol for each system marks the data points from the last 500 frames (60 ns). The red dot marks the corresponding valine centroid distance from the cryo-EM structure. The triangle formed by carbon *β* of V2476 residues from three subunits is shown on the top.

Piezo1 cryo-EM structure and our simulations, the pore's choke point consists of three V2476 from each inner pore helix (Fig. 2e). The centroid distances of three F2480, which correspond to F2754 of Piezo2's lower constriction site, are larger than the ones from V2476 throughout the 1.75 µs, thus are not considered as a choke point here (Fig. S3a). By tracking the size of this choke point using centroid distance from three valine sidechain Cβ atom, we observed an increase of the pore radius in all three systems compared to the original pore size in the cryo-EM structure (Fig. 2e). This crowding-induced pore opening is entirely consistent with our hyperbolic tangent model.

**Anisotropic membrane curvature**. The mismatch between the spherical shape of the Piezo1 dome and the orthorhombic shape of the box induces certain degrees of anisotropic membrane curvature. A hexagonal lattice PBC would be suitable for keeping the threefold symmetry of the channel. However, it is currently not supported by the MD engine in the Anton2 supercomputer, on which we rely on to achieve microseconds AA simulations of Piezo1. To investigate the anisotropic membrane curvature, we fitted the lower leaflets curvature in *xz* and *yz* plane, and plotted against box size along *x*-axis (*x*-dim) and *y*-dim individually (Fig. 3a, b). The box size difference among *box1*, *2*, and *3* were introduced when generating three replicas. Due to lipid fluctuations near the boundary, small variations in *xy*-dimensions are

expected. We notice that the *x*-dim is smaller than *y*-dim in all three systems and thus *x*-dim has more influence on membrane curvature. The smaller the *x*-dim, the smaller the curvature in *xz* plane, thus the flatter the membrane along *x*-axis. The *box1* system, which has the smallest dome–dome distance among three systems (Table S3) shows the smallest membrane curvature in both *xz* and *yz* planes.

**Smaller Piezo–Piezo distances induce larger pore opening**. We next investigated whether the flattening of the membrane corresponded to the flattening of the Piezo1 arms. We calculated *β* (Fig. 3c) between the Piezo1 beam and the Piezo internal axis defined by the center of mass of the cap region and C-terminal domain (CTD) region. We found that the angles of arm2 and arm3 are clearly influenced by the *x*-dim as their principal axes are better aligned with *x*-axis: the smaller *x*-dim, the larger their flattening angle *β*. Because of the domain-swapped homotrimeric arrangement of Piezo1, the arm of each subunit hinges on the pore helices (TM37 and 38) of one neighboring subunit. As a consequence, flattening motions in arms 1, 2, and 3 are likely to influence pore helix conformations in subunits 3, 1, and 2, (i.e., pore 3, 1, and 2), respectively (Fig. 2e). We hence ask whether the flattening of arm2 and arm3 correlates with pore motions, specifically the displacement of V2476, in the inner pore helix of subunits 1 and 2. Figure 3d indicates that the *box1* system, which

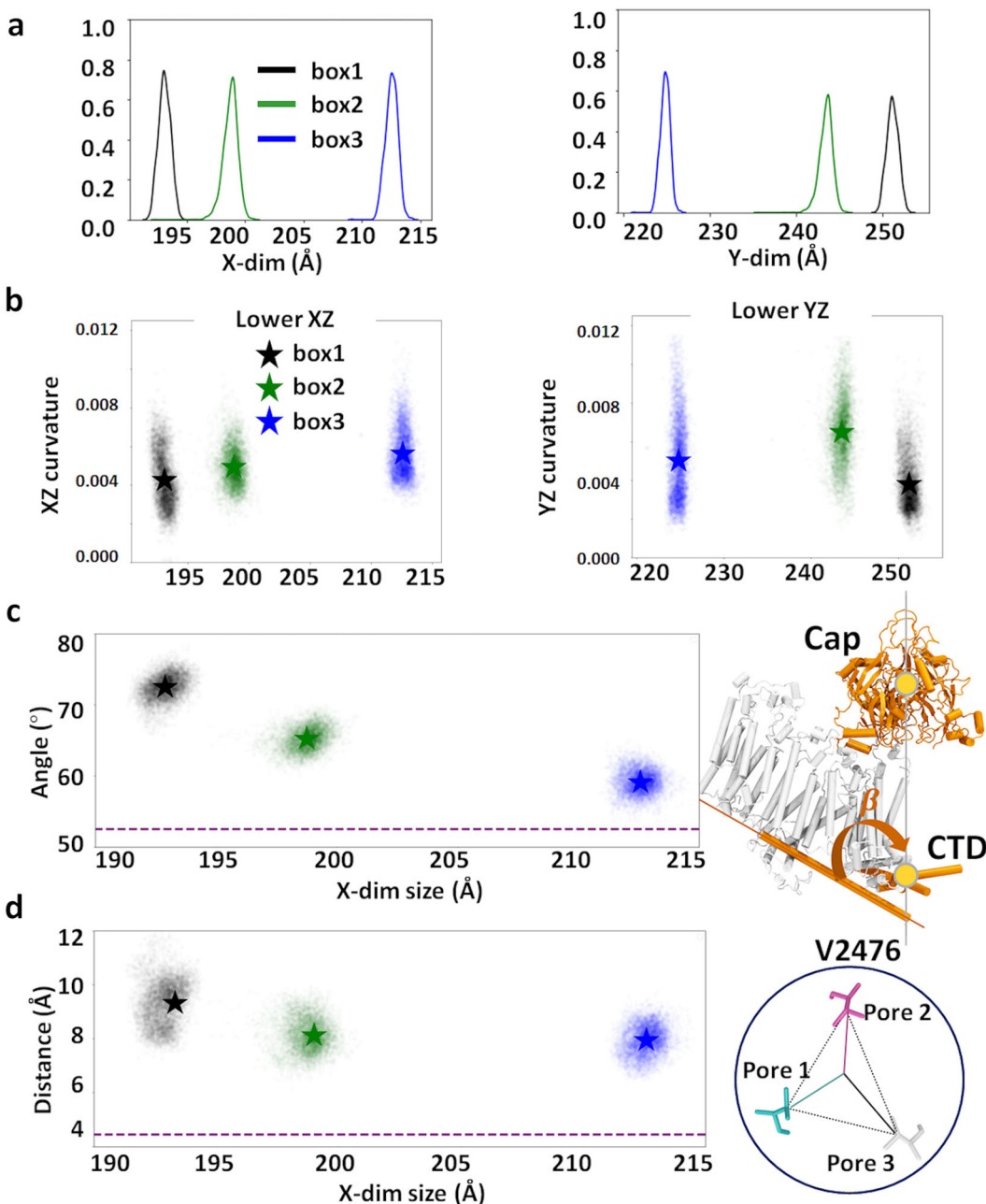

**Fig. 3 Flattening motion of Piezo1 membrane. a** Box sizes along x-axis (x-dim) and y-axis (y-dim) distribution over the last 750 ns of three systems over 1.75 μs. **b** Anisotropic membrane curvature in xz or yz plane, plotted against x-dim or y-dim. The star symbol marks the average values of the last 750 ns trajectory (6250 frames). **c** Flattening angle of the Piezo1 arms plotted against x-dim. The angle β is defined by the angle between each beam, and the Piezo internal axis defined by the COM of cap and CTD region, illustrated on the right. The mean value of arm2 and arm3 from the last 750 ns are plotted here. The star symbol marks the average values. **d** V2476 centroid distances are plotted against x-dim. The pore helix of subunit 1 (pore1) is in contact with arm2 and the pore helix of subunit 2 (pore2) with arm3. The mean value of pore1 and pore2 from the last 750 ns are plotted here. The star symbol marks the average values. In **c** and **d**, the corresponding values in the cryo-EM structure are indicated by purple dashed lines.

has the smallest x-dim, indeed showed the largest centroid distance of V2476.

**Piezo1 conformational changes associated with pore opening.** Since the largest pore size was obtained in *box1* system having the smallest dome–dome distance, we extended this system to 2 μs and evaluated structural changes between 0 and 2 μs. Figure 4a shows that the arms extend in a peripheral direction in addition to their flattening motion component. Zooming into the pore region, an outward tilt of the intracellular end of the inner pore

helices (TM38) enables the widening of the pore at the position of V2476. This tilt rotates the V2476 side chains away from the pore lumen, increasing its diameter (Fig. 4b). The outer pore helices (TM37) have a larger outward motion (Fig. 4b). As a result, the pore radius at the linker region between the cap and TM37 also increases. In addition, we observed ~12° cap domain clockwise rotation (Fig. 4b) and upward displacement of the CTD due to the Piezo dome flattening (Fig. S3b).

A recent study showed that inserting an inter-subunit cysteine bridge between cap residues A2328 and P2382 abolish indentation-evoked Piezo1 currents[17]. The same phenotype was

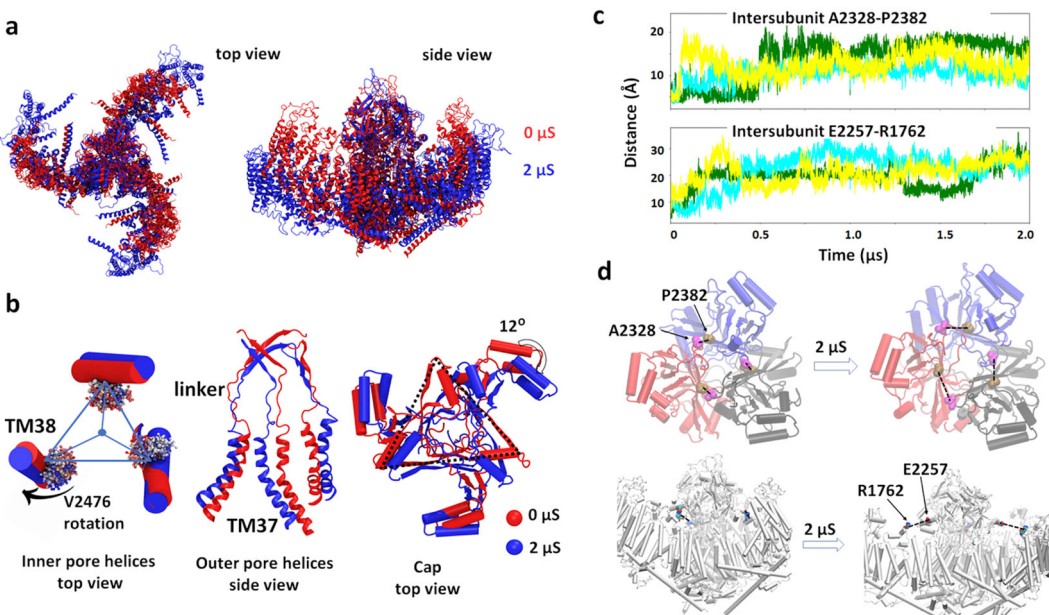

**Fig. 4 Piezo1 conformational changes associated with pore opening. a** Overlap of Piezo1 backbone at 0 μs (red) and 2 μs (blue) of *box1* system; **b** comparing the conformations of Piezo1 inner pore helices (three TM38), outer pore helices (three TM37), and cap domain at 0 μs (red) and 2 μs (blue). The rotation of the hydrophobic gate residue V2476 is illustrated by overlapping the V2476 sidechain trajectories from red to blue. The linker region that connects the cap beta-sheet (residue G2193–G2234) with TM37 is also shown; **c** conformational changes during Piezo1 activation are consistent with inhibitory disulfide bridges. Distance between β carbon of cross-linking residues during the simulation time. Three colors represent three pairs of inter-subunit residues; **d** the increased distances between these inhibitory disulfide bridges illustrated on the protein structure at the beginning and end of the AA simulation.

observed when a disulfide bridge is inserted between the cap residue E2257 and the arm residue R1762. These experimental results indicate that, for both pairs of residues, the inter-residue distance permits disulfide bond formation in the close state, but not in the open state. Our trajectory indeed confirmed that these two inter-residue distances are within disulfide bond formation at the beginning of our simulation and increase beyond disulfide bond formation during the simulation. Briefly, the three inter-subunit A2328–P2382 distances increased from 5 Å to 8–18 Å (Fig. 4c). The A2328–P2382 pairs are located at the base of the cap and linked with the outer pore helices (TM37) through a linker region. The simulation shows that when the arms flatten, the base of the cap widens to enable outward motion TM37 helices (Fig. 4b). This widening cap motion separates A2328 and P2382 beyond disulfide bond formation. In addition, during the flattening of the arms, all three E2257–R1762 distances between cap and arm increased to >20 Å during the trajectory (Fig. 4c, d). In the nonconducting state, cap motions are prohibited by its vicinity with the arms. Thus, the cap rotation shown in Fig. 4b is only enabled upon arm flattening, which separates the cap from the arm. Moreover, both inter-subunit distances are located at the bottom part of the cap, where the upper ionic fenestrations reside (see below Fig. 6a, c).

**Lipids in the pore, pore hydration, and stability**. In the cryo-EM Piezo1 structures, hydrophobic cavities are clearly seen above and below the narrow valine pore constriction. Through these conduits, POPC tails penetrate the pore lumen during our backbone-restrained CG simulations (Fig. S4). Lipids entering the Piezo1 pore during CG simulation has also been reported in a recent preprint[18]. It is currently unknown whether the lipids are present in the nonconducting Piezo1 pore. Such pore occlusion by membrane lipids has been proposed to mediate mechanical gating in mechanosensitive MscS and TRAAK channels[19–22].

However, without further experimental evidence, we do not know whether the pore occlusion by lipids observed here is physiological relevant or an artifact from CG simulation. Our previous 7.9 μs simulation of a reduced Piezo1 with CHARMM36 atomistic force field did not show permanent pore occlusion by lipids in the nonconducting state[9] (Fig. S4). Here, CG simulation is simply used to accelerate the formation of bilayer dome shape. Due to the slow diffusion of atomistic lipids inside the pore, the time needed for these lipids to spontaneously leave the pore may extend well beyond 2 μs. We hence deleted pore lipids at the end of the 2 μs and conducted multiple steps of equilibrium simulations.

We first applied 14.2 mN m$^{-1}$ membrane tension (10 bar lateral pressure) for 50 ns to avoid a sudden membrane tension drop due to lipid removal. The tension was then gradually reduced to zero during 112 ns. During the equilibrium, we found that only the largest pore in *box1* system quickly allowed water and ions to diffuse through the pore and the hydrated pore was stable for the remainder of the simulation (Fig. 5a and pore RMSD in Fig. S5a). In *box2* and *box3* systems, the lipid tails reentered the pore within 50 ns simulation. We suspect that the *box2* and *box3* systems likely represent intermediate states, in which the partially opened hydrophobic gate repels water molecules. It is well known that water molecules inside the channel pore behave differently from bulk water[23–25]. The stability of confined water depends on both the geometry (i.e., radius and symmetricity) and polarity of the pore lumen, hence is channel specific.

**Piezo1 ionic conductance and selectivity in wild-type and E2133Q mutant**. To validate whether the water-conducting pore represents a realistic open state, we calculated the unitary ionic conductance using MD simulations. Constant electric fields corresponding to the transmembrane potentials of −250, −500,

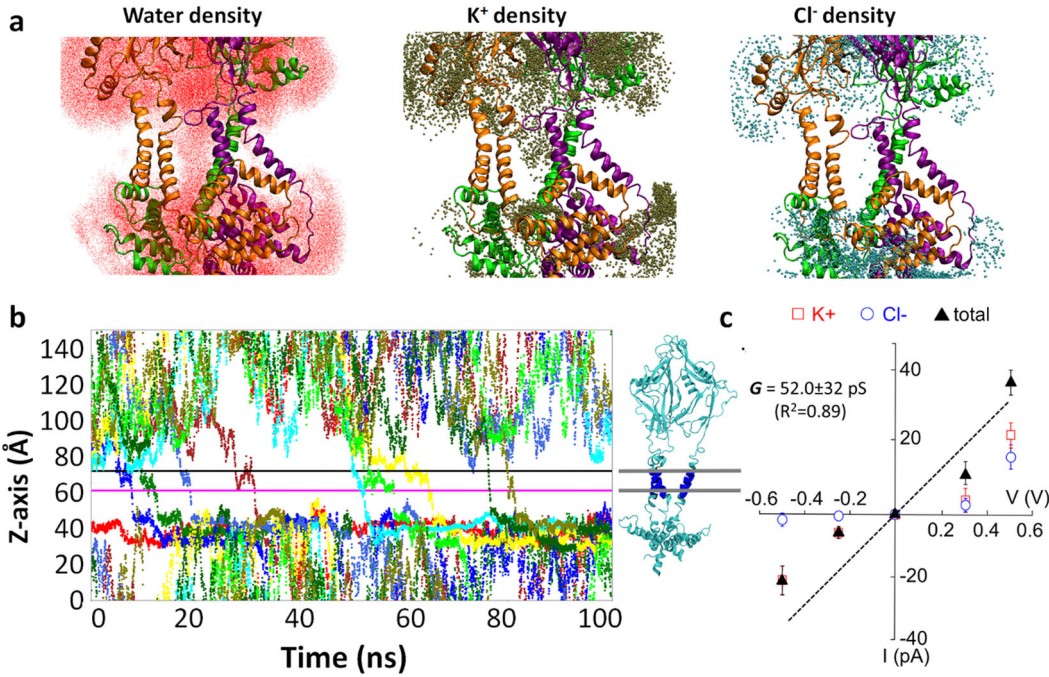

**Fig. 5 Calculated conductance of Piezo1 channel. a** Accumulated water and ion density in the Piezo1 pore from 100 ns simulation at −500 mV voltage. The protein backbone is colored by subunits (orange, purple, and green). Only inner and outer helices, part of cap and CTD domains, are shown for clarity. **b** K$^+$ ion permeation events (nine events) under −500 mV for WT Piezo1 protein. The pore region used to calculate permeation events are defined by the lower and upper boundaries marked by straight lines ($56 < z < 66$ Å), and a cylindrical restraint of radius <35 Å from the central pore axis. **c** The total ionic conductance (black triangles) and individual conductance of K$^+$ (red open rectangles) and Cl$^-$ (blue open circles) ions obtained from MD simulation of WT Piezo1. The black line is a linear fit of the total current with zero voltage zero current restraint (see raw data in Table S4).

+250, and +500 mV were applied perpendicular to the membrane to all the atoms in the simulation box, in the presence of symmetrical 150 mM KCl concentration. The pore remains stable under all tested voltages (Fig. S5b). The ionic conductance computed from the total number of K$^+$ and Cl$^-$ ions transits across the upper and lower boundaries of the pore helix region is 39.7 pS with 85% confidence interval of 22.4–66.8 pS (Fig. 5b and Table S4). Alternatively, calculation of charge displacement along $z$-axis (see cumulative currents in Fig. S6)[26] yields a conductance of 52.0 ± 32 pS (Fig. 5c). Despite the uncertainty in both approaches (see details in "Methods" section) due to the short simulation length and the limitation of the nonpolarizable atomistic force field, the conductance estimated from both approaches are fairly close to the experimentally obtained unitary conductance of 60 pS in the absence of divalent cations[27]. In addition, the Piezo1 pore remains overall cation selective. Because the individual K$^+$ and Cl$^-$ currents exhibit larger fluctuations than the total current, and no Cl$^-$ permeation event was observed under negative voltages within 100 ns, the K$^+$/Cl$^-$ ratio of 2.5–2.9 is likely the lower boundary of Piezo1 cationic selectivity (Fig. 5c). The reported Na$^+$/Cl$^-$ ratio is 7:1 or 13:1 for mouse Piezo1 (refs. [28,29]).

We further tested whether our open state could reproduce the phenotype of a conductance-reducing mutant. The conserved glutamate 2133 located in the anchor region outside of the pore is an important determinant of channel conductance, as charge neutralization mutations E2133A and E2133Q produce a twofold reduction of unitary conductance[28]. Using the stable open-state conformation, we computationally silenced the negative charge of E2133 to mimic the electrostatic effect of E2133Q mutant. After 5 ns equilibrium simulation, ion permeation was measured under 500 mV for 100 ns. This mutant simulation is designed based on the assumption that the conformational perturbation by the

mutated residue is local, in other word, no a large and slow protein conformational change is needed to produce the mutant phenotype. As expected, this charge neutralization reduced the total number of permeation events under 500 mV from 18 events in WT to 8 events in E2133Q mimic mutant (Table S4). Trajectory analysis indicates the E2133Q mimic mutant destabilizes one of the saltbridges with pore residue R2482 (Fig. S7a, b). The absence of R-E saltbridge allows the positively charged R2482 side chains to point down toward the lower fenestrations (Fig. S7c). The resulting repulsive electrostatic force in the K$^+$ permeation pathway likely explained the reduced K$^+$ permeation event observed in the E2133Q mutant simulation (Table S4).

**Multi-fenestrated ion permeation pathway, pore radius, and cation-selective residues.** Our previous MD simulation of a nonconducting Piezo1 model revealed the existence of intracellular cation-selective fenestrations based on the high density of K$^+$ ions near E2487, E2495/2496 underneath the pore region[9]. Here, the K$^+$ permeation pathway captured under a membrane potential not only confirmed these intracellular fenestrations, but also revealed that the EC ions enter the pore via wide lateral fenestrations located between the mouth of the pore and the pore-facing interface of the cap (Fig. 6a and Supplementary Movie 1). The K$^+$ density revealed several hotspots along the ion permeation pathway, indicating longer K$^+$ residence time. The rim surrounding the entryway for EC fenestrations contains the negatively charged residues DEEED 2393-7 (DEEED loop), E2383 in the cap, and D2006 located in the arm of a neighboring subunit. Several negatively charged residues are located along the narrower intracellular entryway, such as E2172 on the anchor, and E2487, E2495/6 on the inner helix (TM38; Fig. 6a). Experimental neutralization of many of these residues

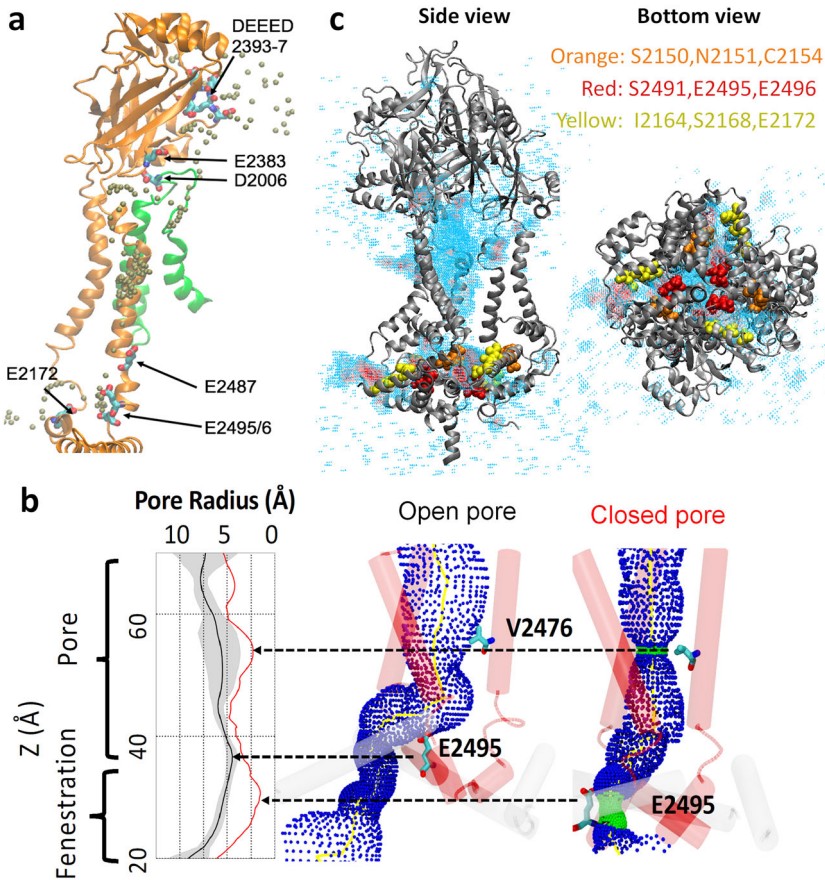

**Fig. 6 Pore radius of Piezo1 channel and multi-fenestration permeation pathway. a** Representative single ion permeation pathway (see Video 1). High K$^+$ density hotspots along the pathway are shown on the protein backbone of a single subunit pore region (orange), except D2006, which is located in the loop of a nearby subunit (green). The K$^+$ is colored in brown. Hotspot residues are shown in licorice with atom color code (red oxygen, cyan carbon, and blue nitrogen). **b** 3D pore radius calculated by the HOLE program[54] using open pore from simulation and nonconducting pore from the initial cryo-EM structure. Only the TM38 backbone is shown for clarity. V2476 and E2495 residues are labeled in licorice. The 2D radius plot shows the average (black line), and standard deviation (gray shade) of pore radius averaged over the last 20 frames (24 ns), and from cryo-EM structure (red line). **c** The side view and bottom view of K$^+$ 3D volmap density near the pore region. The density is color from high (red) to low (cyan) with cutoff = 0.004 Å$^{-3}$. Cap, pore, and CTD regions are shown in newcartoon mode and colored in gray. Nine residues along the lower fenestrations are shown in VDW mode without hydrogens.

(DEEED 2393-7, E2487, and E2495/6) significantly reduced or abolished cation selectivity[28,29], strongly supporting the twisted ion permeation pathway unraveled by our simulations.

Throughout the simulations, the constriction neck (M2493–E2537) at the bottom of the pore remains closed. The three lower fenestrations observed from the ionic pathway start above this constriction neck. When the ions reach the bottom of the TM38 (E2495/6), they exit horizontally between neighboring base helices (L2149–Y2175). The radius profiles connecting the pore with lower fenestrations are shown in Fig. 6b. Comparing with the initial cryo-EM structure, not only the V2476 choke point radius increased from 2.5 to 5.3 Å, but an upward motion at the bottom of TM38 (E2495) also increased the radius of the lower fenestration. Figure 6c shows the accumulated K$^+$ density over 100 ns simulation. The upper and lower fenestrations are clearly shown from the side view and bottom view. Nine residues lining the lower fenestration are highlighted. S2150/N2151/C2154 and I2164/S2168/E2172 are from the anchor region base helices, and S2491/E2495/6 are from the bottom of TM38. A recent functional study reported that mutating those nine residues converts the cation-selective Piezo1 channel into an anion-selective channel[30]. We hence computationally modified the charge of those residues to mimic the electrostatic effect of SNCISESEE-9K mutant without altering covalent bonds and vdW

interactions. Consistent with the functional study, the pCl/pK ratio increased from 0.3 in WT to 10 in 9 K mutant based on the permeation events observed (Table S4).

**Piezo1 contains multiple putative PIP$_2$ binding sites**. Figure S2 shows that PIP$_2$ lipids quickly diffuse toward Piezo1 within 1 μs and remain at the annular region of the protein throughout the 12 μs CG trajectory and the following AA simulations. A large number of PIP$_2$ binding and unbinding events were observed from microseconds CG simulation and AA simulations, which allowed us to identify putative PIP$_2$ binding hotspots in Piezo1. The binding distance is determined by the first minimum from the radial distribution function between arginine/lysine side chains and the PIP$_2$ headgroups (see "Methods" section). Out of 143 cationic residues in each subunit of the current Piezo1 model, only 16 residues interact with at least one PIP2 over 60% of 12 μs CG simulation and over 90% of 2 μs AA simulation time (844, 948/9, 1023-6, 1727/8, 2040/1, 2113, and 2182-5 in Fig. S8a). Those 16 residues are located at three different regions of the intracellular protein surface, namely the peripheral arm region, the convex side of the arm region, and the pore and anchor region (Fig. S8b).

To investigate whether some of those PIP$_2$ bindings are functionally essential, we generated R/K to Q mutations to

neutralize the positive charges and tested the effects of these mutations using electrophysiology. Nearby cationic residues were mutated together to ensure the loss of PIP$_2$ binding at that particular position. At the peripheral arm region (Fig. S8b), we tested a double mutant R884/6Q, which showed a minimal reduction in current amplitudes (Fig. S8c) and excluded the residues on the concave side of the peripheral arm as the missing sequence on the N-terminal region may lead to an overestimation of the PIP$_2$ binding. On the convex side of the Piezo1 arm, PIP$_2$ binding hotspots spread the whole arm region. The double mutant K1201Q/R1204Q showed a minimal decrease in mechanically activated (MA) current amplitudes, while the K1727Q/R1728Q showed a 44% decrease at the maximal stimulation strength (Fig. S8c, d). A single mutant R1023Q at the elbow region of the Piezo1 arms showed MA currents similar to that in wild-type channels; however, the quadruple mutant R1023-6Q showed a current reduction by ~80%. This construct however, showed visibly dimmer GFP fluorescence, therefore some of the decreases were likely due to decreased expression levels. Overall, none of the mutations abolished Piezo1 function, indicating that none of the putative individual binding sites is indispensable for the mechanical activation of Piezo1. It is also consistent with previous results showing that depletion of PIP$_2$ in a cellular context does not entirely abolish Piezo1 activity[15]. None of the mutations changed the inactivation time constant significantly (Fig. S8e).

## Discussion

Simulating Piezo1 in a crowded environment induces global gating motions. Pore opening correlates with an outward motion of outer pore helices, allowing tilting of inner helices and rotation of V2476 side chains away from the pore lumen. In addition, we observed a concomitant rotation and a widening of pore-facing cap sub-domains. The rearrangements of pore-facing subregions of the cap have been recently shown as necessary for Piezo1 activation[17], highlighting the importance of cap flexibility in enabling channel activation. The Piezo1 permeation pathway uncovered here reveals that ions enter the open pore via lateral fenestrations (one per subunit) on both the EC and intracellular sides. Lateral fenestrations are not uncommon as they have been identified in many families of ion channels and transporters. The multi-fenestrated permeation pathway of Piezo1 is also consistent with the observation that the deletion of a C-terminal beam-to-latch region of Piezo1, which forms a cytosolic plug under the central pore, does not yield constitutively open channels[31]. The intracellular lateral fenestration is also entirely consistent with the ion-conducting lateral portal reported recently[30]. In our MD simulation, K$^+$ ions interact with acidic residues (mainly glutamate) known to contribute to ion selectivity both in the EC and intracellular fenestrations, as well as residues that have not yet been experimentally tested. We further validate this ionic pathway using simulated ionic conductance in WT, conductance-reducing mutant E2113Q in the anchor region, and a selectivity-reversing 9 K mutant in the lower fenestration.

Our Piezo1 model and available cryo-EM structures do not contain the disordered loop at the bottom of the Piezo1 from residues 1382–1405, which is termed central plug and lateral plug[30]. The latch region (1406–1420) under the CTD was partially solved in one of the cryo-EM structure (PDB 6BPZ). However, due to the large gap with the rest of the sequence, an additional constraint was necessary to keep this short loop in place during the simulation. Therefore, to avoid the uncertainty introduced by additional restraint, we did not include the latch in our simulation. Interestingly, the Piezo1.1 variant lacking the lateral plug domains exhibits three discrete sub-conductance

levels, suggesting this lateral plug forms three additional activation gates, one for each of the three intracellular fenestrations. Since our Piezo1 model lacks the lateral plug, it is functionally more closely similar to the endogenous Piezo1.1 isoform identified in C2C12 cells[30].

In all available Piezo1 cryo-EM structures, only about two-thirds of each arm are resolved. Hence, the dome size in our current model is necessarily smaller than the dome of a full-length Piezo1. The free energy change associated with the Piezo1 opening transition can be approximated using Eq. (1a), in which $\gamma$ is the membrane tension, $\Delta A$ is the relative change in projected area, $\Delta G_{protein}$ is the free energy of protein conformational change in the absence of membrane tension, and $\Delta G_{memb}$ is the free energy cost of membrane deformations.

$$\Delta G = \Delta G_{protein} - \Delta G_{memb} - \gamma \Delta A \qquad (1a)$$

In single-channel condition (i.e., no membrane footprint overlap), without tension, Piezo is in a nonconducting state ($\Delta G = \Delta G_{protein} - \Delta G_{memb} > 0$), meaning the free energy cost of channel activation is larger than the free energy cost of membrane deformation. In our simulations, the Piezo1 footprint overlap provides additional free energy, $\Delta G_{overlap}$, for deforming the membrane independently of tension:

$$\Delta G = \Delta G_{protein} - (\Delta G_{memb} + \Delta G_{overlap}) - \gamma \Delta A \qquad (1b)$$

The tension-free opening ($\Delta G \leq 0$) thus indicates that $\Delta G_{memb} + \Delta G_{overlap} \geq \Delta G_{protein}$. To induce a 50% open probability without tension, the free energy contribution from the footprint overlap must be in a similar magnitude to the work needed to open a single channel under tension without footprint overlapping, which has been reported to range within 50–150 k$_B$T (ref. [11]).

According to the classic Helfrich–Canham expression, the bending free energy of a homogeneous bilayer $\Delta G_{memb}$ depends on the membrane curvature and bending rigidity. If we approximate Piezo dome as a spherical dome[6,10], $\Delta G_{memb}$ becomes linear to membrane rigidity, independent of the radius of the dome. The shorter arm will certainly decrease the change in Piezo projected area $\Delta A$. But in zero tension condition, $\Delta A$ becomes irrelevant. $\Delta G_{overlap}$ is determined by the distance between neighboring Piezo domes, independent of the size of the dome. Thus, the major influence of the arm length is on $\Delta G_{protein}$, the free energy cost of Piezo1 conformational change. If the Piezo arms are anticipated to act as mechanical levers, the longer the arms, the stronger the output force on the Piezo pore. Therefore, we expect a higher $\Delta G_{protein}$ (more difficult to open) in our current model. In other words, under the same $\Delta G_{memb}$ and $\Delta G_{overlap}$, a full-length Piezo1 model would be expected to open even more readily. It is perhaps not surprising that our first AA Piezo1 model with only 1/3 of each arm (TM25–36,) didn't generate an open state (the truncated model was aimed to enhance the sampling of spontaneous binding of Piezo1 agonist Yoda1)[9]. In this perspective, simulating the full-length arms resolved in the Piezo2 structure[8] will be invaluable for investigating the role of these domains as mechanical levers.

Many independent studies suggest Piezo1 form clusters in cell membranes. Here, using TIRF microscopy, we show that the distribution of the brightness of Piezo1–GFP puncta has a long tail toward higher intensity, consistent with the presence of multiple channels per puncta (Fig. 1a and Fig. S1). Using super-resolution stochastic optical reconstruction microscopy microscopy in the TIRF mode, Ridone et al.[13] measured the area distribution of Piezo1 puncta in cell membranes (~500–6000 nm$^2$, with the most frequent size ranging between 1000 and 1500 nm$^2$). These ranges of values are, on average, larger than the expected projected surface area of an isolated channel at rest, which is

estimated to be ~500 nm² for Piezo1 (ref. [11]; ~700 nm² for the full-length cryo-EM Piezo2 structure[8]). Using patch-clamp electrophysiology, Gottlieb et al.[14] observed that repeated mechanical stimulations eliminate inactivation of all channels present in the patch at once, rather than in a time-dependent manner as one would expect for a stochastic process affecting a population of independent channels. Although the mechanism underlying this collective loss of inactivation remains unclear, this result shows that local stress can affect many channels simultaneously, suggesting they are in close spatial proximity.

Our simulations underscore the exquisite interplay between local membrane geometry and Piezo1 curvature, a property highlighted from cryo-EM and atomic force microscopy studies[10,11]. The clustering effect produced by the PBC of MD simulations mimics a high-density channel cluster and imposes a flattening of the Piezo1-induced membrane footprints, reducing the curvature of the lipid dome and of the Piezo1 arms. This shows that Piezo1 gating is directly governed by membrane topology, which may be manipulated by various mechanical stimuli, as well as the spatial proximity between neighboring channels. This suggests that Piezo1 channels may positively cooperate to gate their pore.

Crowding-induced membrane footprint flattening may not be the only possible mechanism underlying concerted gating in clustered Piezo channels. Cooperative gating may also be governed by direct protein–protein interactions between nearby channels or by indirect interactions mediated by auxiliary proteins[32,33]. Changes in bilayer thickness due to hydrophobic mismatch has been proposed to induce cooperative gating between neighboring MscL mechanosensitive channels and thus may contribute to cooperative gating of nearby Piezo channels. Other entropic contributions due to reduced membrane fluctuations in Piezo clusters may also collectively influence gating property. Whether and how those factors contribute together to the cooperative gating in Piezo clusters with different densities is of interest for further studies.

It should be noted that MD simulations using PBC aimed at studying Piezo1 clustering may lead to several caveats. First, due to the large size of molecular systems simulating Piezo channels and their large membrane footprints, it will be technically challenging to generate a microsecond-long trajectory of a low-density Piezo1 cluster, using available computing resources. Second, channels replicated under PBC condition are mirror images of each other. Thus, MD simulations under PBC cannot replicate spatial heterogeneity of channels in a real cluster. The predicted biphasic behavior of Piezo channels under different cluster densities may be better investigated using high-resolution biophysical approaches.

The presence of PIP₂ binding hotspots is not surprising given the high density of positively charged residues located in the intracellular part of the protein. Most of the hotspots captured here are consistent with PIP₂ binding sites reported elsewhere[34,35]. In addition, the flattening of the Piezo1 arms did not alter PIP₂ density around those hotspots, suggesting PIP₂ binding is independent of channel conformation. It has been shown that the presence of negatively charged PIP₂ or phosphatidylserine in the inner leaflet promote channel activation[15,16,36]. In symmetric PC:PC bilayers, mechanical stress alone is sufficient to activate Piezo1, which indicates that PIP₂ is not an absolute requirement for Piezo1 activation, but rather acts as a modulator[37]. Our mutagenesis data shows that individual neutralization of PIP₂ binding hotspots had minor effects on mechanically induced channel activity, suggesting that a large number of PIP₂–protein interactions might be necessary to collectively shift the thermodynamic equilibrium in favor of the open state.

The apparent interaction of PIP₂ with multiple binding sites on a large protein surface area is in sharp contrast to the binding of PIP₂ to well-defined single binding sites in many PIP₂-regulated ion channels, such as inwardly rectifying K⁺ channels[38], TRPV5 channels[39], and TRPM8 channels[40]. In TRPV5, the PIP₂-bound structure shows an open conformation[39], compared to PIP₂-free structures, indicating that binding of a single PIP₂ molecule per subunit is sufficient to open the channel. Regulation of Piezo1 by PIP₂ through multiple binding sites is likely to be far more complex. Mutations of putative PIP₂ interacting residues had no effect on the inactivation kinetics of MA Piezo1 currents. The quadruple Lysine K2182-85 also showed up on our simulations as a cluster frequently interacting with PIP₂. This cluster is equivalent to residues K2166-69 in the human Piezo1 channel, deletion of which is associated with xerocytosis[41] and displays markedly slower inactivation of MA Piezo1 currents. This may indicate that PIP₂ binding to distinct sites in the channel has different effects on channel function. Since PIP₂ depletion had no significant effect on Piezo1 inactivation[15], it is also possible that the effect of those mutations on channel inactivation is independent of PIP₂ binding. Further studies are needed to understand the complex effect of PIP₂ on Piezo channel activity. It is also possible that PIP₂ amplifies mechanical coupling, which may reduce the entropic cost associated with the conformational rearrangement of the arms, an effect explained by the population-shift theory of allostery[42,43].

In summary, this study shows that the overlap of Piezo1 membrane footprints contributes to channel gating (Fig. 7). The open state generated from MD simulation produces biophysical properties consistent with a large body of published experimental data. The unique multi-fenestrated pore captured from our simulations are supported by experimental neutralization of residues along the ion conduction pathway that reduced or abolished cation selectivity. Mutagenesis study of PIP₂ binding sites identified from multi-scale simulations suggests many lipid–protein interactions cooperate to shift the open/close equilibrium of the channel. This Piezo1 open state model thus provides an avenue for mechanistic investigation of disease mutations and small-molecule drug discovery.

## Methods

**Derivation of hyperbolic tangent membrane footprint model**. The hyperbolic tangent model was chosen to mimic two observed properties of the membrane footprint (code available https://github.com/wesleymsmith/ MembraneFootprintInteractionModel/blob/master/ MembraneFootprintIntersectionModel.ipynb). Firstly, the membrane approaches a flat plateau when distant from the Piezo dome (Fig. 1b). Secondly, the membrane footprint exhibits a "knee" like bend (i.e., a point of maximal concavity/curvature). The sharpness at this knee point grows stronger as the angle of inclination of the arms of Piezo increases. This can be captured in the hyperbolic tangent model by

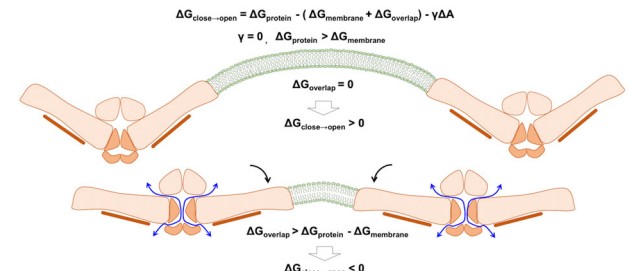

**Fig. 7 Proposed mechanism of crowding-induced Piezo1 channel activation.** The upper panel illustrates two distant Piezo1 channels remain inactive. The bottom panel illustrates two Piezo1 channels in close proximity are activated by crowding-induced arm flattening, revealing a multi-fenestrated ion permeation pathway. Symbols are defined in Eqs. (1a) and (1b).

adding a scaling factor inside the hyperbolic tangent function. More specifically, this scaling constant will be equal to the slope of the hyperbolic tangent model at the origin. Thus, it may be easily related to the angle of inclination of the membrane at the edge of the Piezo dome. If the arms of Piezo have an angle of inclination $\alpha$ with respect to the $xy$ plane, the corresponding slope is given as $\tan^{-1}(\alpha)$. Lastly, if we place two domes a distance of $D$ reduced units apart (note: $D = R/H$), their membrane footprints will intersect at a minimum distance of $D/2$ units apart. Putting this together, we attain the Eq. (2) below:

$$h(x) = \tanh(m_0 \cdot x) \qquad (2)$$

where $h(x)$ is the height of the membrane above the top of the dome at a radial distance of $x$ reduced units away from the edge of the dome, and $m_0$ is the slope of the membrane at the edge of the dome $\tan^{-1}(\alpha)$. Correspondingly, we may calculate the slope of the membrane footprint under this model as in Eq. (3):

$$h'(x) = m_0 \text{sech}^2(m_0 \cdot x) \qquad (3)$$

where $h'(x)$ is the slope of the membrane footprint at position $x$. We may then calculate the slope of the membrane footprint at the point of intersection with another membrane footprint as in Eq. (4):

$$m_{\text{intersect}} = m_0 \text{sech}^2\left(m_0 \cdot \frac{D}{2}\right) \qquad (4)$$

where $m_{\text{intersect}}$ is the slope of the membrane footprint model at the point where it would intersect another membrane footprint when the edges of the two domes are separated by $D$ reduced units. This slope function can be cast as the intersection angle $\theta$ as a function of the angle $\alpha$. To do so, we first note that if the membranes have a slope of $\pm m_{\text{intersect}}$ at their intersection then their corresponding angles are the inverse tangent of their slope. This angle is their effective "angle of inclination" at that point, so the angle formed between them would then be $\pi$ radians minus their sum. This yields Eq. (5):

$$\theta = \pi - 2 \cdot \tan^{-1}(\tan(\alpha)) \cdot \text{sech}^2\left(\tan(\alpha) \cdot \frac{D}{2}\right) \qquad (5)$$

**Summary of system preparation.** Our previous proof-of-concept AA simulation showed that the local membrane curvature imposed by the resting conformation of a truncated Piezo1 takes place over 3 μs (ref. [9]). To reduce computational time, we first conducted a 12 μs CG MD simulation of Piezo1 embedded in a solvated asymmetrical POPC membrane containing 5% PIP$_2$ in the inner leaflet (Table S1). MARTINI CG force field was used to enable rapid lipid diffusion and dome formation, while the Piezo1 backbone was restrained. The channel was aligned along $z$-axis and the bilayer dimension ($xy$) in CG simulation is 698 nm$^2$. We assessed the equilibration of CG MD by monitoring PIP$_2$ lipid density distribution (Fig. S2). The equilibrated CG system was then mapped back to an AA system. Three replicas of ~200 ns AA equilibrium simulations were conducted with different initial velocities without restraint on protein. Assigning different initial velocities (by feeding different random seed numbers) is to make sure each replica of trajectories are different from each other, while the equilibrium velocity distribution follows a Maxwell–Boltzmann distribution at the desired temperature, in this case is the body temperature 310.15 K. After equilibration, each replica was truncated into a smaller box for longer production run with optimal performance on Anton2 supercomputer. The three systems, labeled *box1*, *box2*, and *box3*, have similar membrane surface area and similar lipid distribution between the upper and lower leaflets (Table S2), which indicates similar protein displacement area and similar leaflet tension. Each system ran for 1.75 μs before analysis. System *box1* later was extended to 2 μs before ionic conductance measurement.

**Coarse-grained system preparation.** The CG representation of Piezo1 was constructed from our previously AA Piezo1 model based on the cryo-EM structure (PDB ID 6B3R), which includes residue 782–1365 (Piezo repeat C-F and beam), 1493–1578 (clasp), 1655–1807 (repeat B), and 1952–2546 (repeat A, anchor, TM37, cap, TM38, and CTD)[9]. Based on these atomistic coordinates, the CG model using MARTINI v2.2 force field was obtained through the script *martinize. py* and *insane.py* available from the MARTINI web site[44–46]. For lipids, C16:0/18:1 POPC follows the standard Martini 2.0 lipid definitions and building block rule. A modified PI(4,5)P$_2$ MARTINI model carrying −4 charge was parameterized to be consistent with the experimental data (see the section below). A 28.4 × 24.6 nm membrane bilayer was solvated with explicit water in a simulation box of 28.4 × 24.6 × 25.3 nm. A total of 150 mM NaCl was added to each simulation and kept the whole system charge neutral (hydrated potassium and sodium have the same particle type in CG model).

**Re-parameterization of PI(4,5)P$_2$ Martini force field.** The predominant form of PIP$_2$ in the plasma membrane is PI(4,5)P$_2$ with −4 charge, which indicates one of the phosphate groups being protonated[47]. The current PIP$_2$ model in Martini force field (residue name POP2) is based on PI(3,4)P$_2$ with −5 charge. Hence, the charge on bead name P2 in Martini lipid POP2 is reduced from −2 to −1. Benchmark was done to compare the Martini CG PI(4,5)P$_2$ model with the AA

PI(4,5)P$_2$ structure (residue name SAPI24) in the CHARMM36 lipid force field[48]. Since CHARMM SAPI24 has one more double bond than the Martini POP2 model, the bead name C3A is modified to D3A with its type changing from C3 to C4. PyCGTOOL was used to check the correct CG to AA mapping[49]. The new Martini PI(4,5)P$_2$ model (POP5) is provided in Tables S5 and S6. The comparison of bond length, angle pairs, and radius of gyration of the new PI(4,5)P$_2$ model between the CG and AA are listed in Tables S7–S9. The topology file (itp), pdb file, and a modified INSANE script are provided at https://github.com/reneejiang/cg_pip2_topology_files.

**CG simulation protocol and reverse mapping scheme.** CG simulation in the current study is designed to simply allow faster convergence of membrane topology, while keeping the secondary and tertiary structure of Piezo1 intact. Hence, the protein backbones were kept rigid using positional restraint with a force constant of 1000 kJ mol$^{-1}$ nm$^{-2}$, and an elastic network[50] with a cutoff of 9 Å and a force constant of 500 kJ mol$^{-1}$ nm$^{-2}$. The CG simulation was executed in GROMACS (version 2016.4) simulation package with the standard Martini v2.2 simulation setting[51]. The protein and membrane system were built using a modified enhanced version of the INSANE (INSert membrane) CG building tool. All lipid models and parameters used in this study follow the MARTINI v2.0 lipids, with the addition of the modified Martini PI(4,5)P$_2$ model (POP5). The overall workflow of the simulations includes the initial construction of the Piezo1-embedded membrane, energy minimization, isothermal–isochoric (NVT) and isothermal–isobaric (NPT) equilibration runs, and NPT production runs. Briefly, each system was firstly energy minimized (steepest descent, 5000 steps). NVT simulations were carried out for 0.5 ns at 310.15 K with a timestep of 10 fs. A timestep of 20 fs was used for the following NPT simulations. A cutoff of 1.1 nm was used for calculating both the short-range electrostatic and van der Waals interactions with the Potential-shift-Verlet algorithm applied to smoothly shifting beyond the cutoff. Long-range electrostatic interactions were calculated using the reaction-field algorithm implemented in GROMACS. The neighbor list was updated every 20 steps using a neighbor list cutoff equal to 1.1 nm for van der Waals. The temperature for each group (protein, membrane, ion, and water) was kept constant using the velocity rescale coupling algorithm with 1 ps time constant. For the NPT equilibration step, semi-isotropic pressure coupling was applied using the Berendsen algorithm, with a pressure of 1 bar independently in the cross-section of the membrane and perpendicular to the membrane with the compressibility of $3.0 \times 10^{-4}$ bar$^{-1}$. The pressure in the newly built system was relaxed in a 30 ns simulation using the Berendsen barostat with a relaxation time constant equal to 5.0 ps. Three-dimensional PBC were used. The production step for each system ran for 12 μs using Parrinello–Rahman barostat with a relaxation time constant of 12.0 ps. At the end of each CG simulation, the protein and lipids were mapped into its atomistic representation using Martini backward mapping scheme. The reverse-mapped atomic structure was solvated with CHARMM TIP3P water and 150 mM KCl using the CHARMM36 force field[52].

**AA simulation protocol.** At the end of 12 μs CG MD simulations, the reverse-mapped AA system was truncated to a system with initial size of 21.6 × 22.5 × 16.1 nm$^3$ in $xyz$-dimensions, to reduce the AA system size to ~800,000 atoms. This AA system was first minimized using 50,000 steepest cycles in GROMACS, and then underwent six stages of equilibrium run at 310.15 K using AMBER18 CUDA package, as described in our previous Piezo1 simulation[9]. Three replicas of systems were run on ANTON2 supercomputer with 2.0 fs timestep. Lennard-Jones interactions were truncated at 11–13 Å and long-range electrostatics were evaluated using the $k$-Gaussian Split Ewald method[53]. Pressure regulation was accomplished via the Martyna–Tobias–Klein barostat, to maintain 1 bar of pressure, with a tau (piston time constant) parameter of 0.0416667 ps and reference temperature of 310.15 K. The barostat period was set to the default value of 480 ps per timestep. Temperature control was accomplished via the Nosé–Hoover thermostat with the same tau parameter. The *mts* parameter was set to four timesteps for the barostat control and one timestep for the temperature control. The thermostat interval was set to the default value of 24 ps per timestep. All three replicas at the end of ~200 ns were cut into a smaller size (~710,000 atoms) to take advantage of the high performance on Anton2 for longer production runs (Table S2). In the absence of disordered structure between beam and clasp in the cryo-EM data, a half flat-bottom harmonic restraint with spring constant of 0.12 kcal mol$^{-1}$ Å$^{-1}$ was added between part of the beams (residue 1339–1365) and CTD (residue 2491–2546) to prevent the C-terminal of the beams drifting >30 Å away from the bottom of the pore. The amphiphilic helices (residue 1493–1553) were subjected to RMSD restraints for additional 90 ns at the beginning of Anton2 simulation to ensure the helical structures remain stable for the rest of microseconds production run.

**Gaussian 2D model fitting.** To estimate membrane curvature, the Piezo dome was modeled as a bivariate gaussian fitted to the center of mass coordinates of each leaflet's lipid headgroups (code available https://github.com/wesleymsmith/Piezo_Membrane_Gauss_2D_Model). This gaussian distribution was modeled

using a full covariance matrix formulation taking the form:

$$g(x, zo, h, \mu, C) = zo + h * e^{(x-\mu)^t C^{-1}(x-\mu)} \tag{6}$$

where $C$ is the covariance matrix for the gaussian function, $zo$ is the plateau height of the gaussian dome model, $h$ is distance from the plateau to the peak of the dome model, and $\mu$ is the coordinates of the dome model peak. Since $C$ is a covariance matrix, it is by definition positive semi-definite and symmetric. Therefore, for fitting purposes, this model was recast in terms of the standard deviations $\sigma_{p1}$, $\sigma_{p2}$ of the gaussian model along its two principal axes $p1$, $p2$, and angle of orientation $\theta$ such that

$$C = \left( [p1, p2] \left( R\left[\sigma_{p1}p1, \sigma_{p2}p2\right] \right)^{-1} \right)^{-1} \tag{7}$$

where $R$ is the rotation matrix corresponding to a rotation of $\theta$ radians in the $xy$ plane. This has the added advantage that the two principal curvatures at the peak of the gaussian dome model can be given as $h$ divided by the square of the standard deviation along the corresponding principal axis. The model was then fitted to the headgroup center of mass data for each frame using the python *scipy.optimize* library's minimize function, using the default BFGS method.

For the required initial guess, $zo$ was set to the upper 95th percentile of height values. The height data was then clipped so that the range of the data to be fitted was between the 5th and 95th percentiles of the original height data. This was done to reduce the impact of potential outliers. The value of $h$ was then set to be equal to the range of the fitting data. The value of $\mu$ was set to be equal to the height weighted mean of the headgroup coordinates (where a headgroup coordinate's "height" was computed as its $z$ value minus $zo$). The starting values of $p1$, $p2$, $\sigma_{p1}$, $\sigma_{p2}$, where then generated by treating the heights of the headgroups as effective masses to be used in computing a corresponding inertia tensor matrix. The normalized eigenvectors of this inertia tensor were used as the initial guess for $p1$, $p2$, and their eigenvalues were used for $\sigma_{p1}$, $\sigma_{p2}$. Lastly, the initial value of $\theta$ was determined as the angle of the rotation matrix, which would transform $\left[\sigma_{p1}p1, \sigma_{p2}p2\right]$ into the inertia tensor. During fitting, the value of $h$ was forced to be negative to ensure a concave up dome. The results of this fitting were then used to plot curvature of each leaflet for each of the three membrane configurations (Figs. 2d and 3b).

**In silico ionic conductance measurements**. For each voltage, the current was calculated using two approaches. In the first approach, the ionic currents were computed from the total number of $K^+$ and $Cl^-$ ions permeation events ($N$) over a simulation period ($\tau$), $I = N/\tau$. A permeation event is counted if the ion transits across the upper and lower boundaries of the pore helix region. We call it the boundary-crossing approach. A least-square fitting of the $I$–$V$ curve gave the conductance of 39.7 pS ($R^2 = 0.93$) with 85% confidence interval (CI 85) of 22.4–66.8 pS from Poisson distribution (Table S4). In the second approach, the instantaneous ionic current is computed from charge displacement along $z$-axis, $I(t) = \sum_{i=1}^{n} q_i [z_i(t + \Delta t) - z_i(t)]/\Delta t L$, in which $q_i$ and $z$ coordinate of ion $i$, and $L$ is the length of the channel pore[26]. We call it the $z$-displacement approach. The cumulative current under each voltage is shown in Fig. S6. A least-square fitting of the $I$–$V$ curve gave the conductance of 52.0 ± 32 pS ($R^2 = 0.89$) with the standard deviation estimated from bootstrapping (Fig. 5c and Table S4).

**PIP2 binding site analysis**. The PIP2 binding and unbinding events are counted by GROMACS function "gmx select", which print out whether the atom types PC PL of PIP2 headgroups are within the cutoff distance 5.7 Å of carbon atom "name CZ or CE" connecting with the charged groups of arginine/lysine residues of Piezo1 protein in 2 μs AA simulation trajectory. The cutoff distance is the first minimum distance of the radial distribution function curve calculated between atom types PC PL of PIP2 headgroups and carbon atom name CZ CE of the arginine/lysine residues. Similar calculation is conducted for 12 μs CG simulation trajectory, which prints out whether the bead name PO4 P1 P2 of PIP2 headgroups are within the cutoff distance 6.5 Å of bead name SC1 of arginine/lysine residues connecting to the charged groups. The percent of occupancy is calculated as the total occupancy time divided by the simulation time per each trajectory for each cationic residue.

**Whole-cell patch-clamp electrophysiology**. HEK293 cells were obtained from the American Type Culture Collection (catalog number CRL-1573, RRID: CVCL_0045) and were cultured in minimal essential medium (Life Technologies) containing 10% (v/v) Hyclone characterized fetal bovine serum (Thermo Scientific), and penicillin (100 IU ml$^{-1}$) and streptomycin (100 μg ml$^{-1}$; Life Technologies). Cells were used up to 25–30 passages, when a new batch with low passage number was thawed. All cultured cells were kept in humidity-controlled tissue culture incubator with 5% $CO_2$ at 37 °C. Cells were transiently transfected with cDNA encoding the mouse Piezo1 channel or its mutants tagged with GFP on its N-terminus in the pCDNA3 vector using the Effectene reagent (QIAGEN). Cells were then trypsinized and replated on poly-D-lysine-coated round coverslips 24 h after transfection. Whole-cell patch-clamp recordings were

performed 36–72 h after transfection at room temperature (22°–24 °C), as described previously[15]. Briefly, patch pipettes were prepared from borosilicate glass capillaries (Sutter Instrument) using a P-97 pipette puller (Sutter instrument) and had a resistance of 4–7 MΩ. After forming gigaohm-resistance seals, the whole-cell configuration was established, and the MA currents were measured at a holding voltage of −60 mV, using an Axopatch 200B amplifier (Molecular Devices) and pClamp 10. Currents were filtered at 2 kHz using low-pass Bessel filter of the amplifier and digitized using a Digidata 1440 unit (Molecular Devices). All measurements were performed with EC solution containing 137 mM NaCl, 5 mM KCl, 1 mM $MgCl_2$, 2 mM $CaCl_2$, 10 mM HEPES, and 10 mM glucose (pH adjusted to 7.4 with NaOH). The patch pipette solution contained 140 mM $K^+$ gluconate, 1 mM $MgCl_2$, 0.25 mM GTP, 5 mM EGTA, and 10 mM HEPES (pH adjusted to 7.2 with KOH). Mechanical stimulation was performed using a heat-polished glass pipette (tip diameter, ~3 μm), controlled by a piezo-electric crystal drive (Physik Instrumente) positioned at 60° to the surface of the cover glass, as previously described[15]. The probe was positioned so that 10-μm movement did not visibly contact the cell, but an 11.5-μm stimulus produced an observable membrane deflection. We applied an increasing series of mechanical steps from 12 μm in 0.5-μm increments every 5 s for a stimulus duration of 200 ms. The inactivation kinetics from MA currents were measured by fitting the MA current with an exponential decay function in pClamp, which measured the inactivation time constant (Tau). To calculate this time constant, we used the current evoked by the third stimulation after the threshold in the incrementally increasing step protocol in most experiments, except in cells where only the two largest stimuli evoked a current. In the latter case, we used the current evoked by the largest stimulus, provided it reached 40 pA.

**TIRF microscopy**. HEK293 cells were transfected with GFP-tagged Piezo1 and were plated on poly-D-lysine-coated 25 mm glass coverslips. Measurements were performed 36–48 h after transfection. Culture medium was replaced with the same EC solution used for electrophysiology and cells were mounted on an inverted Ti2E Nikon inverted motorized microscope. Excitation was provided by a 488 nm solid state laser through a 100× NA 1.49 TIRF objective; images were collected with an ORCA Fusion III camera using the Nikon Elements software, and were further analyzed in Image J.

**Statistics and reproducibility**. Electrophysiology data in Fig. S8 are plotted as mean ± SEM and scatter plots using the Origin2019 software, the number of cells indicated for each data point.

**Reporting summary**. Further information on research design is available in the Nature Research Reporting Summary linked to this article.

## Data availability

All simulation input/output files and trajectories are publicly available on Anton2 supercomputer. Source data underlying figures is provided in Supplementary Data 1. Re-parameterized PI(4,5)P2 Martini CG model is available for download at https://github.com/reneejiang/cg_pip2_topology_files.

## Code availability

CG MD simulations were conducted using GROMACS version 2016.4. AA MD simulations were conducted using *pmemd.cuda* version of AMBER18 and a specialized MD engine on Anton2 supercomputer. MD data analysis was conducted using VMD1.9.3 and AmberTools 18 version of CPPTRAJ, and in-house python scripts below. Python3 code for hyperbolic tangent mode is available for download at https://github.com/wesleymsmith/MembraneFootprintInteractionModel. Python3 code for performing Gaussian 2D model fitting of anisotropic membrane curvature is available for download at https://github.com/wesleymsmith/Piezo_Membrane_Gauss_2D_Model.

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

## Acknowledgements

This work was supported by NIH Grants GM130834 (Y.L.L. and J.L.), NS101384 (J.L.), NS055159 (T.R.), GM093290 (T.R.), GM131048 (T.R.), F31-NS100484 (J.S.D.R.), F99-NS113422 (J.S.D.R.), and a WesternU intramural research award (Y.L.L. and J.L.). Computational resources were provided via the Extreme Science and Engineering Discovery Environment (XSEDE) allocation TG-MCB160119 (Y.L.L. and J.L.) and the Pittsburgh Supercomputing Center Anton2 allocations PSCA17006P-18007P (Y.L.L. and J.L.). The XSEDE program is supported by NSF grant number ACI-154862. The Anton2 machine at PSC was generously made available by D.E. Shaw Research, and the Anton2 allocation program at PSC is supported by NIH Grant GM116961.

## Author contributions

Y.L.L, W.J., W.M.B-S., H.Z., and Y-C.L. designed and performed computer simulations; T.R., J.L., J.S.D.R., and S.Z designed and performed experiments; all authors analyzed data; Y.L.L., J.L., and T.R. designed the project and wrote the paper with inputs from all authors.

## Competing interests

The authors declare no competing interests.
