## [Peer Review File · Communications Biology]

Reviewers' comments:

Reviewer #1 (Remarks to the Author):

The manuscript "Crowding-induced Opening of the Mechanosensitive Piezo1 Channel in Silico" by Wenjuan Jiang, John Smith Del Rosario, Wesley Botello-Smith, Siyuan Zhao, Yi-chun Lin, Han Zhang, Jérôme Lacroix, Tibor Rohacs, Yun Lyna Luo reports about a series of coarse-grained (CG) and all-atom (AA) molecular dynamics (MD) simulations of the Piezo1 channel. It documents that the channel's flattening and correlated opening is favored by crowding (along with a theoretical approach). Also, the simulations are in support of PIP2 binding sites along the Piezo1 arms that favor the conformational change (along with a few functional experiments).

The questions concerning gating of the mechanosensitive channel Piezo1 are important and more difficult to address than for other channels that respond to more common stimuli than force. The MD work shown here provides us with an open conformation model that allows us to think how force transduction by means of arm-flattening is translated to the pore region and what the resulting conformational changes in this region may look like. Awaiting an experimental open structure, this is an important step forward for the field.

Thus, I support publication, but I would like the authors to address my questions below.

Line 28: PIP2 binding sites: What was the authors' motivation to address this detailed question? So far researchers have rather been interested in the mechanical bilayer properties?!

Line 51-53: All structures are 0-force structures and therefore expected to be in the closed state. One can even say – given that these structures are solved in detergent micelles and not in a bilayer that has inherent membrane rigidity – that the structures are at "less than 0-force" as has been shown by Lin et al. when the channel is inserted in bilayer without any tension, it will readily be more flattened than the cryo-EM structures.

Line 58-60: Clustering: The argument made that clustering is important is quite important for what is to come. The authors refer to references 11-13 to substantiate that this is the case. Since this is so important for this work, the authors should substantiate what the evidence is in these publications in support of clustering.

Line 64: References 10 and 14 are the same.

Line 87-107: Hyperbolic tangent model:

- The angle alpha (some alphas in the text are α) is defined between the horizontal (putatively flat) membrane plane and the membrane at the beginning of the footprint, just after extending from the channel. Alpha is set modeled between 0 and 90 degrees. However, we know from the structures, that even in detergent micelles, alpha will be 30 degrees or less, because the arms of Piezo enclose roughly 120 degree, thus 60 degrees from the 3-fold axis. Therefore, if the membrane extends from the arms, alpha should in the real case be 30 degree or lower, right?
- The authors plot their hyperbolic tangent model for a series of $D=R/H$ values, but it seems to me that with their simulations (line 131, 2-4nm) they cover only very small R/H ratios?
- Indicated physiologically relevant alphas and computationally tested D in 1b.
- Does the membrane rigidity $K(\text{memb})$ play a role in this theoretical (and the latter computational) considerations?

Line 118: A critical assessment of the used structure, ie incompleteness, detergent micelle, 0-force,

resolution, could/should be mentioned here.

Line 122: "MD simulations ... with different initial velocities": Is this MD-jargon? Is it temperature?

Line 129: "Piezo1 occupies about 32% of the $\sim 480 \text{ nm}^2$ total 129 membrane area": Mention here (see lines 168) that the box has side length of $\sim 22 \text{ nm}$. What does 32% really mean? Is the channel considered a circle in the box, a triangle in the box, a box in the box, or is 32% the molecular surface used as threshold?

Line 131: "the distances between neighboring Piezo arms range 130 from 2 to 4 nm": How is this measured, when I look at panel 2b, I would consider the intermolecular interaction with respect to the membrane deformation that a Piezo1 channel induces as the distance between a circular footprints. When mentioning distances between arms, I fail to get a clear picture because the details of orientation play into it? – see the red drawings on the panel in the attached document.

Line 137 (and comment above): Since $R=2\text{nm}$ corresponds to $D=0.14$, $R=4\text{nm}$ should be just $D=0.28$... Then only the lowest 3 lines in 1b should correspond to MD testable conditions? Given that α is 30 degrees or less for the footprint angle of the structures, one would expect that only interesting effects (critical angle would occur for $D=3$ and higher?

Line 139: Finally, it is mentioned that in real structures α is 30 degrees, should be detailed and discussed much earlier.

Line 144: I can not see or understand panel 2d.

Line 172: Why calling them Box X and Box Y, are these not dimensions X and Y of the same Box?

Line 181: The fact that the membrane flattens more along the X-axis, is this related to the orientation of the trimeric channel in the rectangular box?

Line 203: Is β defined as $90 \text{ degrees} - \alpha$?

Line 207, 215, panels 3c,d: Add dashed lines for cryo-EM structure angle c) and distance d).

Line 228, panel 4b: What is the dashed triangle? It looks shifted?

Line 261: Better?: Does our crowding-induced activated state correspond to a native open state...

Line 263: The results are consistent with the cryo-EM vesicle and HS-AFM flattening studies. The disulfide bridge experiments can only provide a threshold for C-C formation, but there is no structural correspondence to that, nor does it inform us, what is happening beyond that threshold.

Line 274: Can the MD simulations or the hyperbolic tangent model inform us somehow about the quantities needed for flattening the channel and opening. The HS-AFM experiments provided an order of magnitude of spring constant and work to bring about the conformational change. Some functional measurements provide ranges of membrane tension to gate the channel. Can the data here somehow be related to these works?

Line 358, panel 6c, what are the colors in the panel?

Line 371: The binding sites are on the convex side. Did the authors observe in their simulations that

along with flattening, the arms also straightened?

Line 405: The MD simulations faithfully reproduced arm flattening. The functional correspondence with an activated state is more than that.

Line 424-426: Rephrase. Is the conclusions of the authors for selectivity, the electrostatic characteristics of the pore region only? Are there functional/mutagenesis studies in favor of this mechanism?

Reviewer #2 (Remarks to the Author):

The manuscript by Jiang et al. have carried out molecular dynamic simulation studies of the mechanically activated Piezo1 channel and found that Piezo1 clustering and crowding could induce spontaneous channel opening by flattening the peripheral blades. The authors first used the hyperbolic tangent model to fit the geometry of truncated Piezo1 structure that lacking the most distal blade of 12 TMs on the cell membrane, and found that when Piezo1 footprints overlap within a short distance, the Piezo dome structure will be flattened. Then they carried out all-atom MD simulation to show that such flattening can indeed open Piezo1 via assaying the dilation of V2476, which has been considered as a constriction gate in the determined Piezo1 structures. To further show that the simulated flattened structure represents an open conformation, they computationally calculated the single-channel conductance, ion selectivity and simulated the multi-fenestration permeation pathways, and obtained results consistent with published experimental data. Finally, based on their simulation, they also identified three major putative PIP2 interaction regions. However, functional characterization of mutants of these putative PIP2 sites failed to address the functional involvement of these sites in PIP2 regulation of Piezo1. Overall, the simulating result that Piezo1 cluster and crowding can lead to the opening of the channel via flattening the membrane provides novel insights into the gating mechanism of Piezo1. However, there are some major concerns need to be addressed before recommendation for publication.

- 1) The Piezo1 structure used for modeling is an incomplete structure containing only 2/3 of the blade. An incomplete structure will provide inaccurate simulation parameters and lead to inaccurate simulation results. Indeed, the determination of the complete structure of Piezo2 has revealed significant modifications of the size and surface area of the Piezo channel structure (Wang et al., Nature 2019). For instance, the estimated surface area of 480 nm² used in this study might be 700 nm² based on the Piezo2 structure. The previous work from the same group has shown that using a structure containing only 1/3 of the blade failed to simulate the open state. The authors should model the missing 12 TMs of the distal blade and then carry out MD simulation on the complete Piezo1 structure.
- 2) Piezo1 cluster has been showed to be closely related to lipid composition and lipid rafts (Ridone et al. 2020, J. Gen. Physiol). The authors should consider these parameters in their MD simulation.
- 3) On the basis of the current study, membrane topology alone could directly open Piezo1 without applied mechanical force. What will happen if mechanical stimuli are applied in simulation?
- 4) Previous studies have shown that Piezo1-mediated spontaneous Ca²⁺ transients depend on cytoskeleton. So will the crowding-induced Piezo1 opening mechanism indeed occur in native cells?
- 5) The authors considered the determined Piezo1 structure as a fully closed state. However, recent studies of comparing the Piezo1 and Piezo2 structures have suggested that the transmembrane pore of Piezo1 might actually in an open state (at least partial open) compared to the fully closed pore of Piezo2. This has to be considered for the authors to interpret their simulating results. For instance, will their simulation reveal a conformational state of Piezo1 corresponding to the closed state of Piezo2? How about the residue F2480, which correspond to F2754 of Piezo2 as a constriction gate in IH?
- 6) The conformational changes of Piezo1 during the opening process largely resemble the structural

comparison between Piezo1 and Piezo2 shown by Wang et al. Nature 2019. This should be discussed and further highlight the need to properly interpret the conformational state of the Piezo1 structure as pointed out in above point 5.

7) The authors used E2133Q as a mutant to calculate the open pore conductance. However, it is known that E2133 is not located on the transmembrane pore. Please explain how this mutation might allosterically affect the single-channel conductance.

8) In addition to passing through IH-enclosed transmembrane pore, cations have been shown to pass through the intracellular lateral ion-conducting portals equipped with physical lateral plug gates (Geng et al., Neuron 2020). Mutating 9 residues lining the lateral portal convert Piezo1 from cation-selective to anion selective. The authors should highlight whether the simulated lateral ion-conducting pathway in Fig. 6b comprises those residues, and simulate whether mutating those residues to lysine residues can change the conduction of potassium and chloride in the lateral portal.

9) As the author mentioned in the discussion, the previous study has shown that the intracellular lateral portal are physically blocked with a lateral plug gate, which might undergo unplug to open the lateral portal (Geng et al. Neuron 2020). Since that the present simulation lacking this structurally and functionally important lateral plug gate, it is inconclusive for the authors to consider the simulated state as an open state of Piezo1 (please also see above comment point 5).

10) The authors proposed two "selective filters" where K⁺ can be attracted. The use of "selectivity filters" to refer the fenestration sites is not appropriate.

11) Given that no functional and biochemical evidence to verify the putative PIP2 binding sites derived from the simulation, Fig. 7 should be consider as a supplementary figure.

12) In Fig.8, the model needs to be modified to illustrate the crowding effect.

13) There are some Piezo1 molecular dynamic simulation works posted on bioRxiv (Chong et al. 2019, bioRxiv (doi: <http://dx.doi.org/10.1101/783753>), Vecchis et al. 2019, bioRxiv (doi: <http://dx.doi.org/10.1101/823518>). The authors might consider to discuss those studies in the discussion section.

Other minor comments:

14) In Fig 5.C, the x-axis lacks label.

15) In line 341, the amino acid sequence should be DEEED instead of DEED.

16) In line 422, K⁺ ions should interact with acidic residues instead of basic residues.

Decision on manuscript COMMSBIO-20-2046-T

We thank both Piezo channel experts for taking the time to evaluate and comment on our manuscript. We provide here a point-by-point response to all the comments, as well as the new simulation and experimental data to address reviewers questions. We hope that the modifications yield a significantly improved version that merits publication in Communications Biology.

Reviewer #1 (Remarks to the Author):

The manuscript “Crowding-induced Opening of the Mechanosensitive Piezo1 Channel in Silico” by Wenjuan Jiang, John Smith Del Rosario, Wesley Botello-Smith, Siyuan Zhao, Yi-chun Lin, Han Zhang, Jérôme Lacroix, Tibor Rohacs, Yun Lyna Luo reports about a series of coarse-grained (CG) and all-atom (AA) molecular dynamics (MD) simulations of the Piezo1 channel. It documents that the channel’s flattening and correlated opening is favored by crowding (along with a theoretical approach). Also, the simulations are in support of PIP2 binding sites along the Piezo1 arms that favor the conformational change (along with a few functional experiments).

The questions concerning gating of the mechanosensitive channel Piezo1 are important and more difficult to address than for other channels that respond to more common stimuli than force. The MD work shown here provides us with an open conformation model that allows us to think how force transduction by means of arm-flattening is translated to the pore region and what the resulting conformational changes in this region may look like. Awaiting an experimental open structure, this is an important step forward for the field. Thus, I support publication, but I would like the authors to address my questions below.

We greatly appreciate the reviewer's support and suggested improvements

- 1. Line 28: PIP2 binding sites: What was the authors’ motivation to address this detailed question? So far researchers have rather been interested in the mechanical bilayer properties?!**

PIP2 depletion increases the mechanical threshold for Piezo1 activation and thus renders the channel less sensitive to weaker mechanical stimuli (Borbiro et al., 2015). In addition, the existence of PIP2 binding sites has been reported for Piezo2 (Elife 7, e32346, 2018) as well as many other ion channels. These observations, and the large number of positively charged residues at the putative membrane cytoplasm interface (please see the Figure below with R and K residues highlighted in blue licorice) led us to hypothesize that Piezo1 may also possess PIP2 binding site(s). Since we observed consistent PIP2 binding hotspots from multiple simulations, we investigated the functional role of these putative binding sites using functional assays.

Interestingly, none of the tested R/K to Q mutation(s) at PIP2 binding hotspots abolishes mechanically-activated Piezo1 currents. Future studies will be necessary to know whether these sites are functionally important. We have moved the PIP2 results figure to the supporting Figure S8 and discussed the implication of our PIP2 results in the Discussion section.

- 2. Line 51-53: All structures are 0-force structures and therefore expected to be in the closed state. One can even say – given that these structures are solved in detergent micelles and not in a bilayer that has inherent membrane rigidity – that the structures are at “less than 0-force” as has been shown by Lin et al. when the channel is inserted in bilayer without any tension, it will readily be more flattened than the cryo-EM structures.**

This is a very valide point! We agree that the cryo-EM structures solved in detergent micelles are expected to be more curved (less than 0-force) than the channel in bilayer. This is actually consistent with our MD simulations which shows a small flattening of the arms at the beginning of the simulation. To illustrate this, the figure below shows the conformation of our MD simulation at t=0 (pink) and t=200ns (cyan).

Micelles and bilayers differ by their bending rigidity, K_C . Quantitatively, this parameter is related to the free energy cost of changing the membrane curvature, $\Delta G_{\text{membrane}}$, by the Helfrich-Canham equation:

$$\Delta G_{\text{membrane}} = \int_{\text{membrane}} \frac{1}{2} K_C (c_x + c_y)^2 dx dy$$

Where c_x and c_y are the local values of the membrane curvature along x and y axes, which define the bilayer plane. Micelles have a smaller bending rigidity K_C than lipid bilayers, resulting in a smaller $\Delta G_{\text{membrane}}$. This means that it costs less energy for the curved Piezo1 arms to change the local curvature of a micelle than it costs them to change the curvature of a bilayer. This is equivalent to saying that the arms are less flattened in a micelle than in a bilayer.

Besides bending rigidity, temperature also influences protein conformation. Our simulations were conducted at body temperature. Since cryo-EM structures are solved at ultra-low temperatures, one would need to simulate a micelle-Piezo1 system at body temperature and compare it to a bilayer-Piezo1 system at the same temperature to precisely quantify the contribution of bending rigidity on channel conformation.

3. Line 58-60: Clustering: The argument made that clustering is important is quite important for what is to come. The authors refer to references 11-13 to substantiate that this is the case. Since this is so important for this work, the authors should substantiate what the evidence is in these publications in support of clustering.

This is a great suggestion! We briefly describe the actual empirical evidence below; we also added this description to the first results section:

Several different laboratories published that Piezo1 channels tagged with various fluorescent proteins show a non-homogenous, punctate distribution using total internal reflection (TIRF) microscopy, which selectively excites a narrow region of the cell attaching to the glass coverslip, mainly consisting of the plasma membrane (ref 12-14). For the revised manuscript, we performed TIRF experiments, which confirm these earlier data. To illustrate the existence of Piezo1 clusters, we included these data in the revised manuscript, in Figure 1a and FigureS1. In addition, we also discuss the evidence in the literature more carefully in the revised version, as outlined below.

Using TIRF microscopy, Ellefsen et al. (Commun. Bio. 2019), also showed that the amplitude, not the frequency, of Piezo1-dependent calcium flickers increases as acto-myosin traction forces (an endogenous mechanical stimulus of Piezo1) increase, suggesting flickers are produced by Piezo1 clusters rather than isolated channels separated beyond the optical diffraction limit.

Using STORM in TIRF mode, Ridone et al. (JGP 2020) measured the area distribution of Piezo1 puncta in cell membranes, which ranges from 500 to 6000 nm², with the most frequent size ranging between 1000-1500 nm² (the reported spatial resolution of the study is 30 nm). Piezo1 puncta are thus on average larger than the expected projected surface area of an isolated channel at rest, which is estimated to be about 500 nm² for Piezo1 (Yi-Chih Lin et al., Nature 2019) and 700 nm² for the full length cryo-EM Piezo2 structure (Wang et al., Nature 2019).

Using patch-clamp electrophysiology, Gottlieb et al. (Channels 2012) observed that repeated mechanical stimulations eliminates inactivation of all channels present in the patch at once, rather than in a time dependent manner as one would expect for a stochastic process affecting a population of independent channels. Although the mechanism underlying this collective loss of inactivation remains unclear, this result shows that a local stress can affect many channels simultaneously, suggesting they are in close spatial proximity

Line 64: References 10 and 14 are the same.

Deleted the repeated reference. Thanks!

4. Line 87-107: Hyperbolic tangent model:

- **The angle alpha (some alphas in the text are α) is defined between the horizontal (putatively flat) membrane plane and the membrane at the beginning of the footprint, just after extending from the channel. Alpha is set modeled between 0 and 90 degrees. However, we know from the structures, that even in detergent micelles, alpha will be 30 degrees or less, because the arms of Piezo enclose roughly 120 degree, thus 60 degrees from the 3-fold axis. Therefore, if the membrane extends from the arms, alpha should in the real case be 30 degree or lower, right?**
- **The authors plot their hyperbolic tangent model for a series of $D=R/H$ values, but it seems to me that with their simulations (line 131, 2-4nm) they cover only very small R/H ratios?**
- **Indicated physiologically relevant alphas and computationally tested D in 1b.**

We agree with the reviewer. We started this project by asking ourselves “when two dome-shaped channels are in close proximity, will the membrane curvature be more favored or disfavored, compared with a single channel in an infinite membrane?” This hyperbolic tangent model reveals that there is a critical dome-dome distance, below which the flattening is favored and beyond which curvature is favored. This qualitative biphasic relationship is independent of membrane mechanical properties, however, the exact value of this critical distance depends on the geometric parameters of the footprint and bilayer rigidity. Therefore, we kept this hyperbolic tangent model quite general with a wide range of angle alpha and separation coefficient D .

For Piezo1, the initial angle alpha from the cryo-EM structure is 38 degrees (see Fig. 3c dashed line indicate $90-\alpha$). If we estimate the height of the footprint for a single Piezo1 from Figure 2 of Elife 7, e41968, it is on the magnitude of ~10 nm. With our dome-dome distance of 1~3 nm

(see Answer 10b below), the computationally tested D value is 0.1~0.3. Therefore, the separation coefficient D (R/H) in MD simulation is quite small and likely represents the lower limit of Piezo dome separation (i.e., highest cluster density), which is guaranteed to be below the critical dome-dome distance. We have now indicated the physiologically relevant alpha region in blue shaded area (0~38 degrees), and computational tested D value (0.1~0.3) in grey shaded area in Figure 1b. The physiologically relevant D value will depend on the density of Piezo1 clusters on different cell types. Although Piezo1 clusters have been observed by multiple studies (see above), the exact density remains unknown. Therefore, we simulated a wide range of D values from 0.1 to 8.0. If we use 10 nm as the footprint height, a D value of 0.1 to 8.0 corresponds to a dome-dome distance of 1 nm to 80 nm. We have added these Piezo1 relevant values in the Results section “Crowding induces Piezo1 pore opening in three MD simulation replicas”.

5. - Does the membrane rigidity $K(\text{memb})$ play a role in this theoretical (and the latter computational) considerations?

Membrane rigidity is not factored in the hyperbolic tangent model, but is explicitly incorporated in MD simulations.

Incorporating the effects of membrane rigidity in the hyperbolic tangent model would require adding (a) term(s) based on the membrane curvature. Such models were considered in the beginning but were discarded as they generally have no closed-form solution. We discussed at line 443-448 (previous version) that the tangent model revealed a biphasic relationship between Piezo cluster density and membrane curvature, but *“the critical inter-channel distance separating these two scenarios depends on precise geometric parameters of the footprint, bilayer rigidity and membrane tension”*. Because the D values in MD simulations are between 0.1~0.3, Fig 1b shows the footprint overlapping will favor membrane flattening (larger angle theta) as long as the angle alpha is <80 degrees. Since the largest angle alpha is when Piezo is in closed state (38 degrees from cryo-EM structure), it is thus safe to expect a spontaneous flattening of the dome at such high cluster density.

From a mathematical point of view, if this detail is of significant concern (e.g. one wants to take into account membranes of different stiffnesses) an alternative model based upon solutions that ensure zero (or minimal) membrane curvature, e.g. catenoid curves as used in various engineering applications, may be more appropriate. Once again, however, such models would introduce additional parameters that further complicate the simplicity /minimalism which was a goal of this model. We examined other dome-shaped models as well and in general yielded qualitatively similar results. Namely that there is a critical distance at which flattening ceases to reduce the angle of intersection between interacting footprints thus causing bending to become more favorable. This seems to be a general characteristic of models that seek to capture the dome-like shape being modeled. However, no attempt at a rigorous mathematical analysis of what classes of models would yield this critical point phenomenon was made. Such an undertaking may be of general interest in its own right, but such endeavor would likely detract

from the discussion of the larger goals of the overarching investigation and would be better left to a pure mathematics study.

On the other hand, bilayer rigidity is included in the all-atom MD simulation since lipids are explicit and dynamically interacting with the flexible protein. Different bilayer components will have different rigidity. For example, an asymmetric POPC:PIP2 bilayer has a higher bending rigidity than that of a symmetric POPC bilayer (unpublished data from all-atom simulations). Therefore, it is important to keep the bilayer component the same when comparing the membrane footprint overlapping effect. In this study, we used the same POPC/PIP2 bilayer for all three systems, only varying the PBC box size. It is possible to investigate the influence of the bilayer mechanical property on Piezo channel by varying only the lipid composition in MD simulations. However, such investigation must use a substantially larger membrane area to ensure the footprint overlapping effect does not overshadow the bilayer mechanical property effect.

6. Line 118: A critical assessment of the used structure, ie incompleteness, detergent micelle, 0-force, resolution, could/should be mentioned here.

We thank the reviewer for this important suggestion. The following paragraph is added at the beginning of the Results section “Crowding induces Piezo1 pore opening in three MD simulation replicas”.

“The hyperbolic tangent model suggests that for a sufficiently small Piezo-Piezo distance, the overlap of neighboring Piezo1 footprints favor flattening of the Piezo1 dome. We hence explore the influence of the footprint overlap on Piezo conformation using all-atom MD simulations with periodic boundary condition. The atomic model was built using a cryo-EM structure of mouse Piezo1 in a non-conducting (closed) conformation at 3.8 Å resolution (PDB ID 6B3R), solved in detergent micelles in absence of mechanical stimuli. Piezo1 is a large protein with 38 predicted transmembrane helices (TM) per subunit. In this structure, TM1-12 are not present and TM13-16 sidechains are not fully solved. To avoid those structural uncertainties in the atomistic model, we only include TM17-38 for each subunit. Although the arm length in this model is presumably 5/9 of the full length, our simulations show that this model encompasses key structural features allowing Piezo1 to sense membrane topology and open its pore.”

In addition, we also added two new sections in the Discussion to discuss the impact of the incomplete structure on our results: “Missing loops in current Piezo1 model” and “Length of Piezo1 arms”.

7. Line 122: “MD simulations ... with different initial velocities”: Is this MD-jargon? Is it temperature?

To start an MD simulation with a given set of atom coordinates, initial velocity must be assigned to each atom. This is done by assigning random initial velocities to all atoms to ensure that the equilibrium velocity distribution follows a Maxwell-Boltzmann distribution at the desired temperature. For deterministic MD simulation engines (which is the case on the Anton2 supercomputer), if the initial velocities are identical, the trajectory will be identical. Therefore, even though we are conducting all our simulations at 310.15 K (to mimic body temperature), we assign different initial velocities (by feeding different random seed numbers) to make sure each replica of trajectories are different from each other and sample different paths along the free energy landscape. We explained this in the revised Methods section.

- 8. Line 129: “Piezo1 occupies about 32% of the ~480 nm² total membrane area”:
Mention here (see lines 168) that the box has side length of ~22 nm. What does 32% really mean? Is the channel considered a circle in the box, a triangle in the box, a box in the box, or is 32% the molecular surface used as threshold?**

Sorry, this value was not an accurate estimation. We have recalculated the protein occupied area by subtracting the lipid surface area from the total PBC box XY area for upper and lower leaflet individually.

$$Occupancy_{UpperLeaflet} = \frac{BoxArea_{XY} - N_{POPC} * Area_{POPC}}{BoxArea_{XY}}$$

$$Occupancy_{LowerLeaflet} = \frac{BoxArea_{XY} - N_{POPC} * Area_{POPC} - N_{PIP2} * Area_{PIP2}}{BoxArea_{XY}}$$

Using the number of POPC and PIP2 lipids and the box sizes reported in Table S1b, and the area per lipid reported in Charmm36 force field (POPC = 68.3Å², and PIP2 = 67.4Å²), Piezo1 occupies 14.8 ± 0.9% of membrane area in upper leaflet, and 19.8 ± 0.9% in lower leaflet. Standard deviations are obtained from three systems. We have clarified this calculation in the revised Results and Table S1.

- 9. Line 131: “the distances between neighboring Piezo arms range from 2 to 4 nm”:
How is this measured, when I look at panel 2b, I would consider the intermolecular interaction with respect to the membrane deformation that a Piezo1 channel induces as the distance between a circular footprints. When mentioning distances between arms, I fail to get a clear picture because the details of orientation play into it?
– see the red drawings on the panel in the attached document.**

We are sorry for the ambiguity in our description. There are actually two ways to estimate the distance between Piezo proteins.

- a. We calculated the nearest atom distances between neighboring protein backbone (P-P distance), which is a direct measurement of protein-protein distance, regardless of the protein shape (see Figure below)

- b. As the reviewer pointed out, we can also use Piezo dome-dome distance (D-D distance), which seems more relevant to the membrane topology. The radius of the Piezo dome (r) can be estimated using an average centroid distance of a triangle defined by the center of mass of the outer helix in each arm (see orange triangle in figure above). Using centroid distance is necessary because we do not enforce three-fold symmetry during simulations, so the triangle defined by three arms must be a scalene triangle due to thermofluctuation. The average radius is 9.2 nm. So for each system, the D-D distance is the box dimension in X or Y minus $2r$. The table below shows the smallest dome-dome distance along the X dimension, which is between 0.9-2.9 nm. We now use the dome-dome distance for calculation of the separation coefficient D.

System	Box1	Box2	Box3
X (nm)	19.3	19.9	21.3
Dome-dome distance in x (nm)	0.9	1.5	2.9
Y (nm)	25.1	24.4	22.5

Dome-dome distance in y (nm)	6.7	6.0	4.1
-----	-----	-----

10. Line 137 (and comment above): Since $R=2\text{nm}$ corresponds to $D=0.14$, $R=4\text{nm}$ should be just $D=0.28$... Then only the lowest 3 lines in 1b should correspond to MD testable conditions? Given that α is 30 degrees or less for the footprint angle of the structures, one would expect that only interesting effects (critical angle would occur for $D=3$ and higher?

As discussed in question 5 above, in all-atom MD simulations, due to the system size limit on the Anton2 supercomputer, the D values are between 0.1~0.3. The angle α from Piezo1 cryo-EM structure is 38 degrees (calculated using $90-\alpha$, see Figure 3c). According to Figure 1c, this angle corresponds to the critical separation coefficient (D) 1.6~2.8. If we borrow the height of the footprint (H) for a single Piezo1 from Ref (10), it is on the magnitude of ~10 nm. Using these values, the critical distance R is estimated to be 16 ~ 28 nm ($R=D*H$) (Figure 1c), suggesting Piezo1 dome-dome distance below 16 nm will reduce membrane curvature. In current MD simulations, the closest dome-dome distance (1~3 nm) is much smaller than 16 nm, thus it is guaranteed to reduce the membrane curvature. In fact, our simulated Piezo1 cluster likely represents the lower limit of Piezo dome separation (i.e., highest cluster density). The physiologically relevant D value will depend on the density of Piezo1 clusters on different cell types and the exact density is yet to be determined. Therefore, we simulated a wide range of D values from 0.1 to 8.0. If we use 10 nm as the footprint height, a D value of 0.1 to 8.0 corresponds to a dome-dome distance of 1 nm to 80 nm.

11. Line 139: Finally, it is mentioned that in real structures α is 30 degrees, should be detailed and discussed much earlier.

We kept the hyperbolic tangent model very general so that it can be applied to a wide range of channel cluster densities with different separation coefficient D . It is also general for any dome shaped membrane proteins with different α angles. Therefore, the Piezo specific parameters were only mentioned in the next MD simulation section to show that the spontaneous flattening observed from MD simulation is in agreement with the hyperbolic tangent model. The angle α value measured from the initial cryo-EM structure is 38 degrees. The α angle decreases to 25 degrees as the membrane flattens according to the angle β measurement in Fig3c.

12. Line 144: I can not see or understand panel 2d.

To monitor the membrane curvature change during 1.75 μs simulation time, the upper and lower leaflets were modeled as a bivariate gaussian fitted to the center of mass coordinates of the lipid headgroups. The details of the fitting are described in the method section along with our python code. The left figure below is the raw lipid headgroup coordinate data projected in 3D.

The right figure is the top-down xy projection in which the 2D gaussian fitting (color code is the z-coordiante of the center of mass of the lipid headgroup). We now keep the time series of the overall curvature in Figure 2d to show that the membrane curvature decreases in all three systems (the lower the curvature, the flatter the membrane). We have removed the illustration of the 2D gaussian model to avoid confusion.

13. Line 172: Why calling them Box X and Box Y, are these not dimensions X and Y of the same Box?

Thanks for this suggestion. Yes, they are the dimensions X and Y of the same box. To avoid confusion, we now changed Box X and Box Y to x-dim and y-dim. We also changed cut1, cut2, cut3 to box1, box2, and box3 to indicate that they are three different PBC boxes.

14. Line 181: The fact that the membrane flattens more along the X-axis, is this related to the orientation of the trimeric channel in the rectangular box?

Yes. The observation that the membrane flattens more along the X-axis is directly related to the smaller X-dimension of the box, which gives a much closer dome-dome distance along the x-axis than along the y-axis (see the Table in Answer 10). This observation is consistent with our description that the larger the membrane footprint overlap, the flatter the membrane.

15. Line 203: Is beta defined as 90 degrees – alpha?

This is true for cryo-EM structure, but not exactly for MD simulation. Angle beta would be equal to 90-alpha if the channel is perfectly aligned with membrane normal at any time, because alpha in the hyperbolic tangent model is defined as the angle between the arm and xy-plane. However, during MD simulation, protein and membrane are not rigid. Piezo would tilt slightly due to thermal fluctuation of the bilayer and the three arms are not perfectly symmetric at any time point. We therefore define the angle beta as the angle between the Piezo1 beam and the internal axis defined by the center of mass of cap and CTD region.

16. Line 207, 215, panels 3c,d: Add dashed lines for cryo-EM structure angle c) and distance d).

Thanks for this suggestion. Dashed lines added accordingly.

17. Line 228, panel 4b: What is the dashed triangle? It looks shifted?

We added this dashed triangle in Fig. 4b simply to help illustrate the rotation of the cap region from red to blue color. The shift is fixed now.

18. Line 261: Better?: Does our crowding-induced activated state correspond to a native open state...

Thanks for the suggestion. We changed accordingly.

19. Line 263: The results are consistent with the cryo-EM vesicle and HS-AFM flattening studies. The disulfide bridge experiments can only provide a threshold for C-C formation, but there is no structural correspondence to that, nor does it inform us what is happening beyond that threshold.

We agree with the reviewer and we have modified this sentence to:

“In our simulations, the global motion of the Piezo arms upon membrane flattening is consistent with both the cryo-EM vesicles and high-speed atomic force microscopy study. The changes in the atomic distances within the cap and between arm and cap upon membrane flattening is also consistent with the residue-residue distance threshold from disulfide bridge experiments. Does our crowding-induced activated state correspond to a native open state induced by a known mechanical stimulus such as membrane tension? Another important metric available to validate an open state is to determine whether such a model reproduces known ionic conductance and selectivity.”

20. Line 274: Can the MD simulations or the hyperbolic tangent model inform us somehow about the quantities needed for flattening the channel and opening. The HS-AFM experiments provided an order of magnitude of spring constant and work to bring about the conformational change. Some functional measurements provide ranges of membrane tension to gate the channel. Can the data here somehow be related to these works?

We thank the reviewer for encouraging us to expand our discussion beyond the current model.

Gibbs free energy change associated with the Piezo1 opening transition can be approximated using the formula below (γ the membrane tension, ΔA the projected area change with channel opening, $\Delta G_{\text{protein}}$ the free energy of protein conformational change in absence of membrane tension, and ΔG_{memb} the free energy cost of membrane deformations). In a single-channel condition (i.e. no membrane footprint overlap), without tension, Piezo is in a non-conducting state ($\Delta G^{(C \rightarrow O)} = \Delta G_{\text{protein}} - \Delta G_{\text{memb}} > 0$), meaning the free energy cost of protein conformational change is larger than the free energy cost of membrane deformation.

In our MD simulations, the Piezo1 footprint overlap provides additional free energy cost for deforming the membrane regardless of tension. The spontaneous opening without tension ($\Delta\Delta G^{(C\rightarrow O)} < 0$) indicates that $\Delta G_{memb} + \Delta G_{overlap} > \Delta G_{protein}$.

Piezo cluster with membrane footprint overlapping condition:

$$\Delta G^{C\rightarrow O} = \Delta G_{protein} - (\Delta G_{memb} + \Delta G_{overlap}) - \gamma\Delta A$$

If we are allowed to borrow the free energy estimated from HS-AFM, to induce a 50% open probability without tension, the free energy contribution from the footprint overlap must be in a similar magnitude to the work needed to open a single channel under tension (see below for detail).

At a single channel condition with 50% open probability,

$$\Delta G^{C\rightarrow O} = \Delta G_{protein} - \Delta G_{memb} - \gamma\Delta A = 0$$

According to the HS-AFM experiments, it takes about 50 kT to flatten a full length Piezo1 in reconstituted vesicles. Hence,

$$\Delta G_{protein} - \Delta G_{memb} \approx 50kT$$

To induce to 50% open probability of a Piezo using membrane footprint overlapping instead of tension,

$$\Delta G_{overlap} = \Delta G_{protein} - \Delta G_{memb} \approx 50 kT$$

We have added this discussion in the revised version.

21. Line 358, panel 6c, what are the colors in the panel?

To improve the visualization of ionic density in fenestration, we have replaced Figure 6c with a 3D density of potassium ions. We also labelled the nine residues along our fenestration to show that this simulation predicted pathway is entirely in agreement with the lateral portal reported recently by Geng et al Neuron 2020, PMID: 32142647.

22. Line 371: The binding sites are on the convex side. Did the authors observe in their simulations that along with flattening, the arms also straightened?

Indeed, Piezo arm straightening is accompanied by arm flattening (see below, white is 6B3R structure with PIP2 binding residues in one of the arm highlighted in blue, cyan is the structure after simulation with the same PIP2 binding residues in pink). One can imagine that the position/orientation of the PIP2 binding residues may change upon arm straightening. This is precisely the reason we compared PIP2 binding residues in both coarse-grained simulation (in which the protein backbone conformation is constrained) and in all-atom simulation (in which protein is flexible). As seen in Figure S8, the top binding residues (i.e., hotspots) are fairly consistent between two simulations, which indicate the PIP2 binding sites are not altered by arm straightening and the binding is predominantly governed by electrostatics. It is also encouraging that the PIP2 binding spots captured by other reported CG simulations are largely overlapping with ours (bioRxiv 10.1101/787531, Chong et al bioRxiv 10.1101/783753) .

23. Line 405: The MD simulations faithfully reproduced arm flattening. The functional correspondence with an activated state is more than that.

We have deleted this misleading sentence.

24. Line 424-426: Rephrase. Is the conclusions of the authors for selectivity, the electrostatic characteristics of the pore region only? Are there functional/mutagenesis studies in favor of this mechanism?

Yes, we have cited “experimental neutralization of these negatively charged residues (2393-7, E2487, E2495/6) significantly reduced or abolished cation selectivity (26, 27)” (line 344-346 in previous version).

In addition, as reviewer2 pointed out in question 8, a recent study by Geng et al. (Neuron 2020) provided the first experimental evidence of intracellular lateral fenestrations that we observed from our voltage simulations. We have made a new Figure 6c illustrating that our conductance pathway is entirely consistent with the lateral portal in Geng et al., Neuron 2020. To further validate our lower fenestration, we conducted new simulation of SNCISESEE-9K mutant mimic, as suggested by reviewer2. Consistent with Geng’s functional results, we observed that the 9K mutant converts Piezo1 from cationic-selective to anion-selective. The new result is added in the Results section “Multi-fenestrated ion permeation pathway and cation-selective residues”.

Reviewer #2 (Remarks to the Author):

The manuscript by Jiang et al. have carried out molecular dynamic simulation studies of the mechanically activated Piezo1 channel and found that Piezo1 clustering and crowding could induce spontaneous channel opening by flattening the peripheral blades. The authors first used the hyperbolic tangent model to fit the geometry of the truncated Piezo1 structure that lacking the most distal blade of 12 TMs on the cell membrane and found that when Piezo1 footprints overlap within a short distance, the Piezo dome structure will be flattened. Then they carried out all-atom MD simulation to show that such flattening can indeed open Piezo1 via assaying the dilation of V2476, which has been considered as a constriction gate in the determined Piezo1 structures. To further show that the simulated flattened structure represents an open conformation, they computationally calculated the single-channel conductance, ion selectivity and simulated the multi-fenestration permeation pathways, and obtained results consistent with published experimental data. Finally, based on their simulation, they also identified three major putative PIP2 interaction regions. However, functional characterization of mutants of these putative PIP2 sites failed to address the functional involvement of these sites in PIP2 regulation of Piezo1. Overall, the simulating result that Piezo1 cluster and crowding can lead to the opening of the channel via flattening the membrane provides novel

insights into the gating mechanism of Piezo1. However, there are some major concerns need to be addressed before recommendation for publication.

1) The Piezo1 structure used for modeling is an incomplete structure containing only 2/3 of the blade. An incomplete structure will provide inaccurate simulation parameters and lead to inaccurate simulation results. Indeed, the determination of the complete structure of Piezo2 has revealed significant modifications of the size and surface area of the Piezo channel structure (Wang et al., Nature 2019). For instance, the estimated surface area of 480 nm² used in this study might be 700 nm² based on the Piezo2 structure. The previous work from the same group has shown that using a structure containing only 1/3 of the blade failed to simulate the open state. The authors should model the missing 12 TMs of the distal blade and then carry out MD simulation on the complete Piezo1 structure.

We thank the reviewer for pointing out the concern about using truncated Piezo1 structure in MD simulations. While we agree that simulating a full-length model would be better, we would like to clarify that performing a full-length Piezo1 all-atom MD simulation is problematic for the following reasons: (1) there is no full length Piezo1 structure currently available (2) number of atoms in a full-length model may exceed the limits on the Anton2 supercomputer, and (3) the addition of the N-terminal repeats is not expected to change our observation that the overlap of Piezo footprints provides additional free energy to flatten the arms and open the pore.

1. There is currently no full-length Piezo1 structure. In all three high-resolution cryo-EM Piezo1 structures, only two-thirds of each arm are resolved. To model a full Piezo1 structure, we would need to build a homology model based on the structure of the Piezo2 homolog. This operation would add >500 residues based on the homology model. This large addition will inevitably introduce large uncertainty about the structure and organization of these missing domains (Piezo repeats). Hence, such a homology model may not correspond to the native full-length structure. The only “full-length” Piezo1 homology model reported so far is a coarse-grained models (bioRxiv 10.1101/783753). The coarse-grained model is in general suitable to study local interactions between a protein (with constrained protein conformation) and its surrounding lipids. However, to capture the protein conformational change that leads to an open pore, an all-atom model and long-timescale simulation are both necessary. We stated previously (line 432-434) that a future simulation of the full-length Piezo2 seems a much better use of computational resources.
2. Anton2 has a maximum limit of ~700,000 atoms, which may be exceeded when simulating a full-length Piezo1 homology model (<https://www.psc.edu/anton-rfp>). While it is possible to simulate a full-length Piezo1 on a regular computer cluster, the timescale of simulation will be reduced to few hundreds of nanoseconds (e.g. one microsecond will take about 500 days on a common GTX980 GPU card), a full order of magnitude smaller than the timescale reported here. Our three replicas of Piezo1 simulations range between 1.75 to 2 microseconds (made possible by Anton2 supercomputer). Our data

indicate, the development of a stable open state takes more than 1 microsecond. In other words, the crowding-induced opening will be completely missed if we simulate a full-length Piezo1 on an inevitably shorter timescale.

3. A full-length simulation will unlikely change our conclusion, namely that neighboring dome-shaped channels can influence each other in a cooperative manner by modulating the local membrane topology. First, the biphasic relationship between Piezo dome-dome distance and membrane flattening is independent of the size of the dome (not factored in the mathematical hyperbolic tangent model). Second, there are several physical reasons that explain why shorter arms (smaller dome) still allow Piezo1 to sense membrane topology and open its pore in MD simulations. Shorter Piezo arms reduce the size of the Piezo dome, which are expected to alter all the terms on the right side of the equation below (please see detail in answer 20 to reviewer 1). We discuss each term below:

$$\Delta\Delta G^{O\rightarrow C} = \Delta G_{protein} - (\Delta G_{memb} + \Delta G_{overlap}) - \gamma\Delta A$$

- 1.1. The change in the Piezo projected area ΔA will be smaller upon activation. But in zero tension condition, ΔA becomes irrelevant.
- 1.2. According to the classic Helfrich-Canham expression, the bending free energy of a homogeneous bilayer $\Delta G_{membrane}$ depends on the membrane curvature and bending rigidity (please see detail in answer 2 to reviewer 1). If the Piezo dome is approximately $\frac{1}{3}$ of a sphere, $\Delta G_{membrane}$ becomes approximately 8.4 folds the membrane rigidity, independent of the radius of the dome.

$$\Delta G_{membrane} = \int_{membrane} \frac{1}{2} K_C (c_x + c_y)^2 dx dy$$

$$c_x = c_y = 1/R$$

$$\Delta G_{membrane} = \frac{1}{3} \int_S \frac{1}{2} K_C \left(\frac{2}{R}\right)^2 dA = 8.4 K_C$$

- 1.3. $\Delta G_{overlap}$ is determined by the distance between neighboring Piezo domes (see Figure1 hyperbolic tangent model), thus independent of the size of the dome.
- 1.4. $\Delta G_{protein}$ is the free energy cost of Piezo conformational change from a closed pore to an open pore. If the Piezo arms are anticipated to act as mechanical levers, the longer the arms, the stronger the output force on the Piezo pore. Therefore, we expect a higher $\Delta G_{protein}$ (more difficult to open) in our current model. In other words, under the same $\Delta G_{membrane}$ and $\Delta G_{overlap}$, a full-length Piezo1 model would be expected to open even more readily. It is thus perhaps not too surprising that in our previous MD simulation of a truncated Piezo1 model with only $\frac{1}{3}$ of the arm length, $\Delta G_{protein}$ was too high to allow pore opening.

In order to emphasize and clarify the incompleteness of the current Piezo1 model, we have added the following paragraph at the beginning of the MD simulations Result section:

“The hyperbolic tangent model suggests that at a sufficiently small Piezo-Piezo distance, the overlap of neighboring Piezo1 footprints will favor the flattening of the Piezo1 dome. We hence explore the influence of the footprint overlapping on Piezo conformation using all-atom MD simulations with periodic boundary conditions. The atomic model was built using a cryo-EM structure of mouse Piezo1 in closed conformation at 3.8 Å resolution (PDB ID 6B3R), solved in the detergent micelle without mechanical stimuli. Piezo1 channel is a large protein with 38 predicted transmembrane helices (TM) per subunit. In this structure, TM1-12 are not present and the side chains of TM13-16 are not fully solved. To avoid those structural uncertainties in the atomistic model, we only include TM17-38 for each subunit. Although the arm length in this model is presumably 5/9 of the full length, our simulations show that this model encompasses key structural features allowing Piezo1 to sense membrane topology and open its pore.”

In addition, we also added two new sections in the Discussion to discuss extensively the impact of the incomplete structure on our results: “Missing loops in current Piezo1 model” and “Length of Piezo1 arms”.

2) Piezo1 cluster has been showed to be closely related to lipid composition and lipid rafts (Ridone et al. 2020, J. Gen. Physiol). The authors should consider these parameters in their MD simulation.

The direct outcome of Piezo1’s membrane footprint overlap is the change of the membrane topology, specifically, the curvature of the membrane. This result does in no way exclude the role of lipid composition, including lipid rafts. In fact, our ultimate goal is to study the force-from-lipid mechanism on the Piezo channels in different lipid compositions. We believe the result reported here is a critical step forward in how to investigate the role of lipid composition on Piezo conformational changes using MD simulation.

Ridone and others have shown that depleting cholesterol or disrupting cholesterol organization slowed both the activation and inactivation of Piezo1 in response to membrane stretch. Using super-resolution STORM imaging, they showed that in the plasma membrane, cholesterol has some effect on the size of the Piezo clusters, albeit not as a key determinant (Fig.5 of Ridone et al. 2020, JGP). Even though the Piezo clustering in plasma membrane may depend on cholesterol and/or lipid rafts, in MD simulations we can simply mimic a highly dense Piezo1 cluster using periodic boundary conditions. While we discussed (previous version line 435-466) that such a “virtual cluster” has its own limitation, the membrane footprint overlap-induced Piezo opening is consistent with Ridone et al’s observation. Our study strongly supports the importance of Piezo1 clusters because it requires much less work to activate Piezo1 in a cluster than as isolated channels.

It is of great interest to simulate Piezo1 in a realistic plasma membrane model. For example, a coarse-grained Piezo1 model has been simulated in a multi-component bilayer model (bioRxiv 10.1101/792564 and 10.1101/783753) or in a complex plasma membrane model containing >60 lipids (bioRxiv 10.1101/787531) to study lipid interaction and distribution around Piezo1. However, to capture the protein conformational change that leads to an open pore, a fully

flexible atomistic model and long-timescale simulation are necessary. It is often tempting to simulate Piezo1 in its most realistic environment possible, but in reality, the results can often be compromised due to the following limitations in an all-atom MD simulations:

1. The more lipid types we use for the bilayer, the longer simulation time is needed for the convergence of lateral-distribution of each lipid type. As we showed in Fig. S2, it took at least 2 μ s for the PIP2 lateral distribution to stabilize in a POPC bilayer in a coarse-grained model. All-atom lipids diffuse slower than coarse-grained lipids.
2. The risk of having leaflet tension mismatch in the complex asymmetric bilayer is higher (Biophys J. 2018, PMID: 30297133).
3. The molecular mechanism of force-from-lipids can be quite complex. As illustrated below, some lipids may enhance force-from-lipids by increasing the coupled motion between protein and lipids (because the force from lipids is never equally distributed on the protein surface), changing bilayer mechanical properties, or stabilizing the open state, or increasing the Piezo clustering. It would be difficult, if not possible, to pinpoint the contribution from each lipid type when simulated in a complex bilayer.

With these caveats in mind, we chose to start our investigation with a simple binary lipid composition (POPC/PIP2) as a cautious first step. An important lesson brought up by this study is that in order to investigate Piezo channels in different lipid compositions, a substantially larger bilayer is needed to minimize the Piezo footprint overlap so that the contribution of $\Delta G_{overlap}$ does not overshadow the other free energy contributions in Piezo1 activation. Cholesterol, fatty acid, and sphingomyelin also have an impact on Piezo inactivation kinetics. The Piezo open state model reported here will allow us to study the role of lipid in the inactivation process.

3) On the basis of the current study, membrane topology alone could directly open Piezo1 without applied mechanical force. What will happen if mechanical stimuli are applied in simulation?

It is of great interest to study the Piezo1 response to various mechanical stimuli, such as poking, suction, membrane stretch, or shear flow. It is currently unclear whether those stimuli activate Piezo1 through a common mechanism. In the current study, we found that the Piezo1 footprint overlapping serves as one of the mechanical stimuli that flatten the membrane without tension,

which suggests that when Piezo1 channels are in closer proximity, the mechanical threshold to activate them is lower than when they are further apart.

In the current simulation of a highly dense Piezo1 cluster, spontaneous opening of the pore suggests that the free energy contribution from the simulated footprint overlapping may be in a similar magnitude to the work needed to open a single channel under tension. Thus no additional tension is needed to maintain a stable open pore (see detail in answer 20 for reviewer 1). When we further add membrane tension, the pore will be stretched up to a point that the protein will unfold, thus the RMSD of the pore will not be stable (see Figure below). Without a stable conformation, one cannot claim that a true state has been reached. Even if a stable state is reached with a water-conducting pore, it is still important to compare the calculated ionic conductance with experimental values to confirm that it is not just an artificially stretched pore by excess tension.

4) Previous studies have shown that Piezo1-mediated spontaneous Ca²⁺ transients depend on cytoskeleton. So will the crowding-induced Piezo1 opening mechanism indeed occur in native cells?

The acto-myosin cytoskeleton is required for the cell to create endogenous traction forces. Inhibition of myosin activity by blebbistatin indeed abolishes spontaneous Piezo1-dependent calcium flickers (Pathak et al., PNAS 2014). This is simply because the endogenous mechanical stimulus can not be generated in absence of cytoskeleton.

Piezo1 channels (both endogenous and heterologously expressed) are preferentially located in dense clusters (Ellefsen et al., Commun. Bio. 2019, Ridone et al, JGP 2020, and Gottlieb et al., Channels 2012), see also our new data in Figure 1a and supplemental figure 1. Although the exact density (number of channels per surface area) in these clusters remains to be determined, it is quite likely that some degree of footprint overlap in the Piezo1 clusters exist in a cellular environment. According to our predictions, the mechanical threshold for activation should be lower in a cluster than for an isolated channel, therefore it may well be that Piezo1 footprint overlap is the factor that allows Piezo1 opening by relatively weak forces exerted by the cytoskeleton and thus induce Ca²⁺ flickers. We have included a similar discussion in the revised manuscript.

Exogenous stimuli on the other hand can open Piezo1 channels regardless of the presence of the cytoskeleton. It is now well established that Piezo1 activation does not require cytoskeletal elements *per se*, as it is purely gated by the force from lipids (e.g. Syeda et al., Cell Reports 2016 PMID: 27829145, and eLife 2015 PMID: 26001275, Romero et al., Nat. Comm. 2019, PMID: 31435011). However, cytoskeleton disruption decreases the apparent elasticity of the membrane (stress/strain ratio) and prevents long range force transmission across the cell. These two consequences lead to two opposite effects on electrophysiology recordings: in whole-cell configuration, channel activity is diminished because the force is dampened by the membrane in absence of cytoskeleton, leading to a smaller area being effectively stimulated by poking. In patch configuration, decreased elasticity means the membrane stretches more for a given pressure pulse, decreasing the threshold for activation (see Nourse and Pathak, Semin. Cell Dev. Biol. 2017, PMID: 28676421).

5) The authors considered the determined Piezo1 structure as a fully closed state. However, recent studies of comparing the Piezo1 and Piezo2 structures have suggested that the transmembrane pore of Piezo1 might actually in an open state (at least partial open) compared to the fully closed pore of Piezo2. This has to be considered for the authors to interpret their simulating results. For instance, will their simulation reveal a conformational state of Piezo1 corresponding to the closed state of Piezo2? How about the residue F2480, which correspond to F2754 of Piezo2 as a constriction gate in IH?

Yes, the impressive full-length Piezo2 structure brought up a key difference in the pore region. In the Piezo1 structure, there is only a lower transmembrane constriction site, while in Piezo2, the upper transmembrane constriction site is also closed. It is possible that the upper constriction site also exists in Piezo1, but was not captured by the Piezo1 static structure or MD simulation. Therefore, we cannot rule out the possibility that the Piezo1 structure is partially open, even though from reviewer1's point of view, the Piezo1 captured in detergent micelle might be a "less than 0 tension" closed state (overly closed state). At this point, we do not want to over-speculate how a true closed state should look like. In addition, Geng et al. Neuron 2020 have recently shown convincing evidence that the intracellular fenestration of Piezo1 can be sterically blocked by a disordered loop (called lateral plug gate), which is not present in any cryo-EM structures or all-atom model of Piezo1. Therefore, it is not accurate to call current Piezo1 model as a closed state model. Compared with the Piezo2 structure, the upper part of Piezo1 pore is dilated, rendering a funnel shaped pore with the minimum radius of 2 Å at the lower constriction site. We and other group's MD simulations have shown that the lower constriction site is too narrow to conduct ion and water (*ref 9*, bioRxiv 10.1101/823518), corresponding to a non-conducting state of Piezo1. For this reason, we have changed the word "closed" to "non-conducting" pore for describing our initial structure as we show the lower constriction site has to open substantially to allow water/ion permeation. The upper constriction site may be influenced by the cap position captured in the cryo-EM condition. Future simulations of both Piezo1 and Piezo2 under the same condition might help resolve this interesting puzzle.

Regardless of the uncertainty in the upper constriction site, our simulations show that the current Piezo1 model is able to sense the force from membrane fluttering and open the lower

constriction site. During our simulations, we monitored several pore residues, including F2480, as potential constriction sites according to the static structures (PDB 6bpz, 6b3r, 5z10). Molecular dynamics revealed that F2480 side chains are flexible and rotate out of the pore region. The cation-pi interaction with K2479 is broken. Thus F2480 does not constitute the constriction as V2476 (see Figure below, added in Figure S3). Thus we only reported V2476 as the lower constriction site, which only opens upon membrane flattening. We have modified the introduction (page 3) and results (page 9) section to reflect the points above.

6) The conformational changes of Piezo1 during the opening process largely resemble the structural comparison between Piezo1 and Piezo2 shown by Wang et al. Nature 2019. This should be discussed and further highlight the need to properly interpret the conformational state of the Piezo1 structure as pointed out in above point 5.

Indeed, there are many features captured from our Piezo1 simulations that are in agreement with the structural comparison between Piezo1 and Piezo2.

1. The ionic conductance pathway shows that neither the extracellular central cavity of the cap nor the intracellular constriction neck is involved in ion permeation.

2. While the existence of the upper transmembrane gate in Piezo1 needs further investigation, our simulation clearly shows that the lower transmembrane gate can open upon arm flattening. This is the first “dynamic” confirmation of the mechanosensitive function of curved Piezo arms, which was originally proposed by Ge et al. Nature 2015 for Piezo1 and now by Wang et al. Nature 2019 for Piezo2.

3. In addition to the central pore, we observed ion enter/exit through three intracellular fenestration sites (also called lateral portals), which are clearly shown in the Piezo2 structure also. The residues E2495/6 in the intracellular vestibule connecting pore and fenestrations correspond to E2769/79 in Piezo2. Mutating these residues affects ion permeation in Piezo1 and Piezo2 (Zhao et al. Neuron 89).

4. It is interesting that the cap domain of Piezo1 structure (6B3R) twists in a clockwise direction relative to Piezo2 (viewed extracellularly). In our simulation, we observed that, when the pore region is aligned for each time frame, the cap twists further in a clockwise direction upon Piezo1 activation (Fig.4b) and the upper portion of the pore opens even wider (Fig.6b). This seems to suggest that Piezo2 is in a more closed conformation than the Piezo1 structure. Since the cap is not in contact with the membrane, the cap rotation during our simulations is likely a result of the Piezo arm flattening.

5. Compared with the Piezo2 structure, the Piezo1 structure (6B3R) shows slightly flatter arms/beams and upward displacement of the CTD. The CTD moved further upward upon Piezo1 arm flattening (see below the cap and CTD position, red is from 6B3R, blue is after simulation, residue V2476 and E2495 are shown in VDW, added as Figure S3b). This is consistent with the observation described by Wang et al that Piezo2 is in a more closed conformation.

It is our hope that with what we learned from Piezo1 simulations so far, we will ultimately be able to conduct a rigorous simulation of the Piezo2 full-length model. Further MD simulation of Piezo2 can help explore whether the cap rotation will open the upper constriction site in Piezo2 and whether the full-length arms will further facilitate mechanosensation. We added the description of the dilated Piezo1 pore compared with Piezo2 in the Introduction. We also added the upward CTD motion in Figure S3 and discussed the motion of cap and CTD domain in comparison with Piezo2 structure on page 24.

7) The authors used E2133Q as a mutant to calculate the open pore conductance. However, it is known that E2133 is not located on the transmembrane pore. Please explain how this mutation might allosterically affect the single-channel conductance.

Although there are many mutations along the pore and fenestration that are worth testing, we found this E2133 mutation especially interesting because it is not located in the pore nor in a fenestration. In general, reproducing the phenotype of pore mutation is fairly straightforward because the mutated side chains often electrostatically interact with the ions or sterically block their journey through the pore. But to reproduce the phenotype of an allosteric mutation, located outside the pore, will require the channel conformation/dynamics to be accurate beyond the pore. Thus, within our limited computational resources, we chose to test E2133.

The conductance-reducing phenotype of E2133A/Q/K strongly suggests an electrostatic contribution. The closest positive residue (likely to interact with the native E2133) in the pore is R2482. Plotting the E-R distance throughout the trajectory indicates that, among three E-R saltbridges, at least one pair remains stable (distance < 4Å) for 1.75 μ s (green line in panel a in the figure below). By contrast, the E2133Q mutation breaks this salt bridge within 40ns (panel b below). The absence of R-E saltbridge allows the positively charged R2482 sidechains to point down towards the lower fenestrations. The panel c below shows the distance between R2482 sidechain and the fenestration (we define fenestration as the center of mass of 9 residues reported by Geng et al., Neuron 2020, illustrated on the right). The resulting repulsive electrostatic force in the K⁺ permeation pathway likely explained the reduced K⁺ permeation event in the E2133Q mutant. As a result, the total number of permeation events under 500 mV from 18 events in WT to 8 events in E2133Q mimic mutant. We have added this observation in Results section (page 18-19) together with Figure S7. We also stated in the revised version that our mutant simulation is designed based on the assumption that the conformational change induced by mutant is local, in other word, no a large and slow protein conformational change is needed to produce the mutant phenotype.

8) In addition to passing through IH-enclosed transmembrane pore, cations have been shown to pass through the intracellular lateral ion-conducting portals equipped with physical lateral plug gates (Geng et al., Neuron 2020). Mutating 9 residues lining the lateral portal convert Piezo1 from cation-selective to anion selective. The authors should highlight whether the simulated lateral ion-conducting pathway in Fig. 6b comprises those residues, and simulate whether mutating those residues to lysine residues can change the conduction of potassium and chloride in the lateral portal.

We would like to thank the reviewer for the excellent suggestion! We were indeed quite excited to see the results from Geng et al., Neuron 2020 coming out towards the end of completing our project. Their study provided the first experimental evidence of intracellular lateral fenestrations that we observed from our voltage simulations. We have made a new Figure 6c (shown below) illustrating that our conductance pathway is entirely consistent with the lateral portal in Geng et al., Neuron 2020.

To further validate our lower fenestration, we conducted new simulation of SNCISESEE-9K mutant mimic, as suggested by reviewer. Consistent with Geng's functional results, we observed that the 9K mutant converts Piezo1 from cationic-selective to anion-selective (the pCl/pK ratio increased from 0.3 in WT to 10 in 9K mutant based on the permeation events reported in Table S2). The new result is added in the Results section "Multi-fenestrated ion permeation pathway and cation-selective residues".

9) As the author mentioned in the discussion, the previous study has shown that the intracellular lateral portal are physically blocked with a lateral plug gate, which might undergo unplug to open the lateral portal (Geng et al. Neuron 2020). Since that the present simulation lacking this structurally and functionally important lateral plug gate, it is inconclusive for the authors to consider the simulated state as an open state of Piezo1 (please also see above comment point 5).

Our Piezo1 model as well as available cryo-EM structures do not contain the disordered loop at the bottom of the Piezo1 from 1382 to 1405 (termed central plug and lateral plug). The latch region (1406-1420) under the CTD was partially solved in one of the cryo-EM structure (PDB 6BPZ). However, due to the large gap with the rest of the sequence, additional constraint will be necessary to keep this short loop in place during the long simulation. We hence chose to not include the latch in our model to avoid additional uncertainty in the simulation. Interestingly, a Piezo1 variant Piezo1.1 structure lacking the density for the lateral plug domain, has 1.4 fold of conductance of Piezo1 and about twice the mechanosensitivity of Piezo1. Piezo1-(Δ 1382-1420) or Piezo1-(Δ 1382-1445) also have comparable increase in conductance. Those data strongly suggest the lateral plug may physically block the fenestration. Since our Piezo1 model lacks the lateral plug, it is functionally comparable to the endogenous Piezo1.1 isoform (which lacks the lateral plug) identified in C2C12 cells by Geng et al. We have discussed this important point by adding a new discussion section “missing loops in current Piezo1 model”.

10) The authors proposed two “selective filters” where K⁺ can be attracted. The use of “selectivity filters” to refer the fenestration sites is not appropriate.

We changed to “cationic selective sites” instead of “selectivity filters”.

11) Given that no functional and biochemical evidence to verify the putative PIP2 binding sites derived from the simulation, Fig. 7 should be consider as a supplementary figure.

Thanks for the suggestion. We have now moved Fig.7 to supplementary figure S8.

12) In Fig.8, the model needs to be modified to illustrate the crowding effect.

Great suggestion. We have modified Figure 8 (now Figure 7) accordingly to emphasize the footprint overlapping.

13) There are some Piezo1 molecular dynamic simulation works posted on bioRxiv (Chong et al. 2019, bioRxiv (doi: <http://dx.doi.org/10.1101/783753>), Vecchis et al. 2019, bioRxiv (doi:<http://dx.doi.org/10.1101/823518>)). The authors might consider to discuss those studies in the discussion section.

There are four bioRxiv articles describing Piezo1 simulation. Coarse-grained simulation using Martini force field has been an efficient tool to study membrane-protein interactions if the protein does not need to undergo a conformational change. There are three coarse-grained (CG) simulations of Piezo1 reported so far. Buyan et al (bioRxiv 10.1101/787531) conducted a CG simulation of a full-length Piezo1 in a complex plasma membrane model, thus revealed enrichment and depletion of specific lipids around the protein. Another group has used CG simulations to study the binding of sphingomyelin/ceramide (Shi et al. bioRxiv 10.1101/792564) and cholesterol/PIP2 (Chong et al bioRxiv 10.1101/783753). In this study, we also take advantage of this approach to accelerate the convergence of membrane curvature and the lateral distribution of PIP2. It is encouraging that CG simulations conducted by different research groups thus far all captured a similar Piezo dome shape and the PIP2 binding sites identified by CG simulations are largely overlapping as well.

The major difference in the current study is that we are aiming at exploring whether the Piezo1 arm flattening will generate a fully open pore that captures realistic ionic conductance and permeation pathway. For this task, we converted the equilibrated CG model back to AA model to allow protein conformational change in response to membrane flattening. Because of the smaller membrane patch used in AA simulation, we found the membrane footprint overlapping alone is sufficient to flatten the Piezo1 arms and the opened pore was stable for 2 us. Interestingly, Vecchis et al (bioRxiv 10.1101/823518) simulated an all-atom Piezo1 model under different membrane tension and found the pore can open (within 100 ns) when tension is at least 67.8 mN/m. Although it remains to be tested whether the pore under tension allow ion permeation in a similar fashion as we observed, the relation between arm flattening and pore opening is indeed consistent with our observation.

We have cited them in the results and discussion.

Other minor comments:

We thank reviewer for thoroughly reading and assessing our manuscript.

14) In Fig 5.C, the x-axis lacks label.

We have added the missing x-axis label as the reviewer pointed out.

15) In line 341, the amino acid sequence should be DEEED instead of DEED.

Modified sequence as DEEED.

16) In line 422, K⁺ ions should interact with acidic residues instead of basic residues.

We have changed to acidic residues.

REVIEWERS' COMMENTS:

Reviewer #1 (Remarks to the Author):

The authors have provided detailed responses to my questions and undertaken according changes in the manuscript. I think the manuscript is much more clear now, and am favorable to move forward towards publication.

Reviewer #2 (Remarks to the Author):

The authors have addressed the raised comments. This reviewer supports the publication of this paper.